# CTCF orchestrates the germinal centre transcriptional program and prevents premature plasma cell differentiation

Arantxa Pérez-García[1,*,†], Ester Marina-Zárate[1,*], Ángel F. Álvarez-Prado[1], Jose M. Ligos[2], Niels Galjart[3] & Almudena R. Ramiro[1]

In germinal centres (GC) mature B cells undergo intense proliferation and immunoglobulin gene modification before they differentiate into memory B cells or long-lived plasma cells (PC). GC B-cell-to-PC transition involves a major transcriptional switch that promotes a halt in cell proliferation and the production of secreted immunoglobulins. Here we show that the CCCTC-binding factor (CTCF) is required for the GC reaction *in vivo*, whereas *in vitro* the requirement for CTCF is not universal and instead depends on the pathways used for B-cell activation. CTCF maintains the GC transcriptional programme, allows a high proliferation rate, and represses the expression of Blimp-1, the master regulator of PC differentiation. Restoration of Blimp-1 levels partially rescues the proliferation defect of CTCF-deficient B cells. Thus, our data reveal an essential function of CTCF in maintaining the GC transcriptional programme and preventing premature PC differentiation.

[1] B Cell Biology Lab, Vascular Pathophysiology Area, Fundacion Centro Nacional de Investigaciones Cardiovasculares Carlos III, Melchor Fernández Almagro 3, 28029 Madrid, Spain. [2] Cellomics Unit, Cell & Developmental Biology Area, Fundacion Centro Nacional de Investigaciones Cardiovasculares Carlos III, Melchor Fernández Almagro 3, 28029 Madrid, Spain. [3] Department of Cell Biology and Genetics, Erasmus Medical Center, PO Box 2040, 3000 California Rotterdam, Netherlands. * These authors contributed equally to this work. † Present address: Epigenetics of Cancer and Ageing Group, Cancer Research UK Beatson Institute, Garscube Estate, Switchback Road, Bearsden, Glasgow G61 1BD, UK. Correspondence and requests for materials should be addressed to A.R.R. (email: aramiro@cnic.es).

Germinal centres (GC) are microstructures that develop in secondary lymphoid organs as a result of B-cell activation by antigen and that allow the generation of high-affinity memory B cells or long-lived antibody secreting plasma cells (PC), the effector cells of the humoral immune response[1,2]. After antigen engagement, naive B cells are activated by interaction with CD4$^+$ T cells and initiate a vigorous proliferative response that promotes the clonal expansion of the cells that recognized the antigen. Proliferating GC B cells engage in the somatic remodelling of immunoglobulin (Ig) genes by somatic hypermutation, which introduces mutations in the variable region of the immunoglobulin genes and generates clonally related B cells expressing immunoglobulins with slightly altered binding specificities[1,3]. Within these closely related clones, only those B cells with a higher affinity for the initiating antigen are selected for survival and further proliferation in the process known as affinity maturation[2]. Thus, the biology of GCs is extremely complex and entails proliferation, B-cell receptor signalling for survival, cell death and cell fate decisions along with a significant reorganization of the genomic architecture that encodes the GC B-cell transcriptome[4].

The exit of B cells from the GC and their differentiation into PCs involves a major transcriptional switch that promotes on one hand, a halt in cell-cycle progression and immunoglobulin diversification, and on the other, a boost in the transcription of immunoglobulin genes together with a massive production of secreted immunoglobulin[5]. Two important transcriptional regulators orchestrate the transition from naive to GC B cell and from GC B cell to PC: Bcl-6 and Blimp-1. The transcriptional repressor Bcl-6 is considered the master regulator of the GC reaction. Bcl-6 is upregulated at the GC stage and regulates the expression of genes involved in B-cell activation, survival, DNA-damage response and cell-cycle arrest, among other pathways. Mice lacking Bcl-6 cannot form GCs or produce high-affinity antibodies (reviewed in ref. 6). Blimp-1 is a transcriptional regulator expressed at the transition from GC to PC differentiation. B cells that lack Blimp-1 are unable to proceed to the PC fate and cannot secrete immunoglobulins[7]. Blimp-1 acts as a transcriptional repressor that promotes B-cell proliferation arrest, establishes the PC transcriptional programme and triggers immunoglobulin secretion[7–10]. Importantly, Bcl-6 and Blimp-1 establish mutual negative regulatory loops, such that Bcl-6 prevents Blimp-1 expression and Blimp-1 is considered key to extinguish the GC reaction[8,11,12]. In this regard, the GC and PC differentiation stages can be considered as antagonistic transcriptional programs orchestrated by Bcl-6 and Blimp-1.

The CCCTC-binding factor (CTCF) is a ubiquitous architectural protein with eleven zinc-finger domains. Although initially described as a transcriptional regulator of the c-myc proto-oncogene[13–15] that establishes physical barriers on the DNA acting as a transcriptional insulator[14], studies have shown that CTCF is also associated with regions of active transcription[16]. CTCF mediates long-range chromatin loops to facilitate or prevent promoter–enhancer interactions[17–19], suggesting that CTCF may have a general function in the control of gene transcription (reviewed in ref. 20). A number of studies have addressed the function of CTCF during B-cell development. Removal of CTCF-binding sites at the immunoglobulin heavy chain locus has revealed an important function of CTCF in the regulation of V(D)J recombination during bone marrow differentiation. In addition, elimination of CTCF in early B-cell precursors, although compatible with immunoglobulin heavy chain recombination, resulted in a block in B-cell differentiation in the bone marrow[21–29]. However, the function of CTCF in mature B cells, and particularly during the GC reaction, is unclear.

Here we use a conditional mouse model to deplete CTCF specifically in GC B cells. We find that CTCF is required for the GC reaction in vivo. By contrast, the sensitivity to CTCF deficiency in vitro is selectively dependent on the pathways mediating B-cell activation. Extensive transcriptome and functional analyses reveal that CTCF is required to maintain the GC transcriptional programme and to prevent premature PC differentiation by facilitating cell proliferation and by blocking Blimp-1 expression. Thus our data unveil CTCF as an important regulatory factor for late B-cell differentiation.

## Results

**CTCF is required for the germinal centre reaction in vivo.** To address the role of CTCF during the GC reaction we specifically depleted CTCF in GC B cells by breeding mice carrying a floxed CTCF allele with transgenic AID-CRE mice (AID-CRE$^{TG/+}$), where the Cre recombinase is inserted together with a cDNA encoding the human CD2 molecule (hCD2) in a BAC construct that contains the complete activation-induced deaminase (AID) locus (Supplementary Fig. 1a). AID is a deaminase expressed in GC B cells that initiates antibody diversification by somatic hypermutation and class switch recombination. Thus, in CTCF$^{fl/fl}$; AID-CRE$^{TG/+}$ mice the expression of CRE and the deletion of CTCF is triggered by the expression of AID—therefore specifically expressed in GC B cells—and surface expression of hCD2 can be used to track GC B cells and CTCF depletion (Supplementary Fig. 1b). As expected, analysis of B-cell differentiation in bone marrow and spleen did not show any difference between CTCF$^{fl/fl}$; AID-CRE$^{TG/+}$ and CTCF$^{fl/+}$; AID-CRE$^{TG/+}$ mice (Supplementary Fig. 1c–e and Supplementary Table 1), indicating that B-cell differentiation is not affected before GC formation in CTCF$^{fl/fl}$; AID-CRE$^{TG/+}$ mice.

To determine the role of CTCF in GCs, we immunized CTCF$^{fl/fl}$; AID-CRE$^{TG/+}$ (CTCF deficient), and CTCF$^{fl/+}$; AID-CRE$^{TG/+}$, CTCF$^{+/+}$; AID-CRE$^{TG/+}$ CTCF$^{+/+}$; AID-CRE$^{+/+}$ (control) mice with sheep red blood cells (SRBC) and analysed the GC response in spleen 7 days later (Fig. 1a–c). We found that the proportions of GC B cells (Fas$^+$GL7$^+$) and IgG1$^+$-switched cells were markedly reduced in CTCF$^{fl/fl}$; AID-CRE$^{TG/+}$ mice when compared to all groups of control mice (Fig. 1a,b). Accordingly, the expression of hCD2 is almost undetectable in CTCF$^{fl/fl}$;AID-CRE$^{TG/+}$ mice (Fig. 1a,b). The absence of GC B cells upon CTCF depletion was also evident by immunofluorescence staining, which showed that immunized CTCF$^{fl/fl}$ mice have normal splenic and follicular architecture but completely lack PNA+, GC B cells (Fig. 1c). Given that we did not detect any significant difference among the three control groups of mice, hereafter all experiments will be performed using CTCF$^{fl/+}$; AID-CRE$^{TG/+}$ (CTCF$^{fl/+}$ for brevity) and CTCF$^{fl/fl}$; AID-CRE$^{TG/+}$ (CTCF$^{fl/fl}$) littermates as control and CTCF-deficient groups, respectively. Consistently with the novo GC formation, we found a profound GC block in Peyer's patches (Supplementary Fig. 1f). Together, these results show that CTCF is absolutely required for the GC reaction in vivo.

**B-cell activation pathway determines CTCF loss sensitivity.** To address the molecular mechanisms of GC impairment in CTCF-deficient mice we made use of a standard in vitro system in which LPS/IL4 stimulation of naive B cells recapitulates many features of the GC reaction such as induction of B-cell proliferation, AID expression and class switch recombination. Splenic B cells from CTCF$^{fl/fl}$ or CTCF$^{fl/+}$ mice were stimulated with LPS/IL4 and we used the expression of the hCD2 reporter as a surrogate to track the extent of B-cell activation. Surprisingly, we did not find any difference in the proportion of hCD2 positive

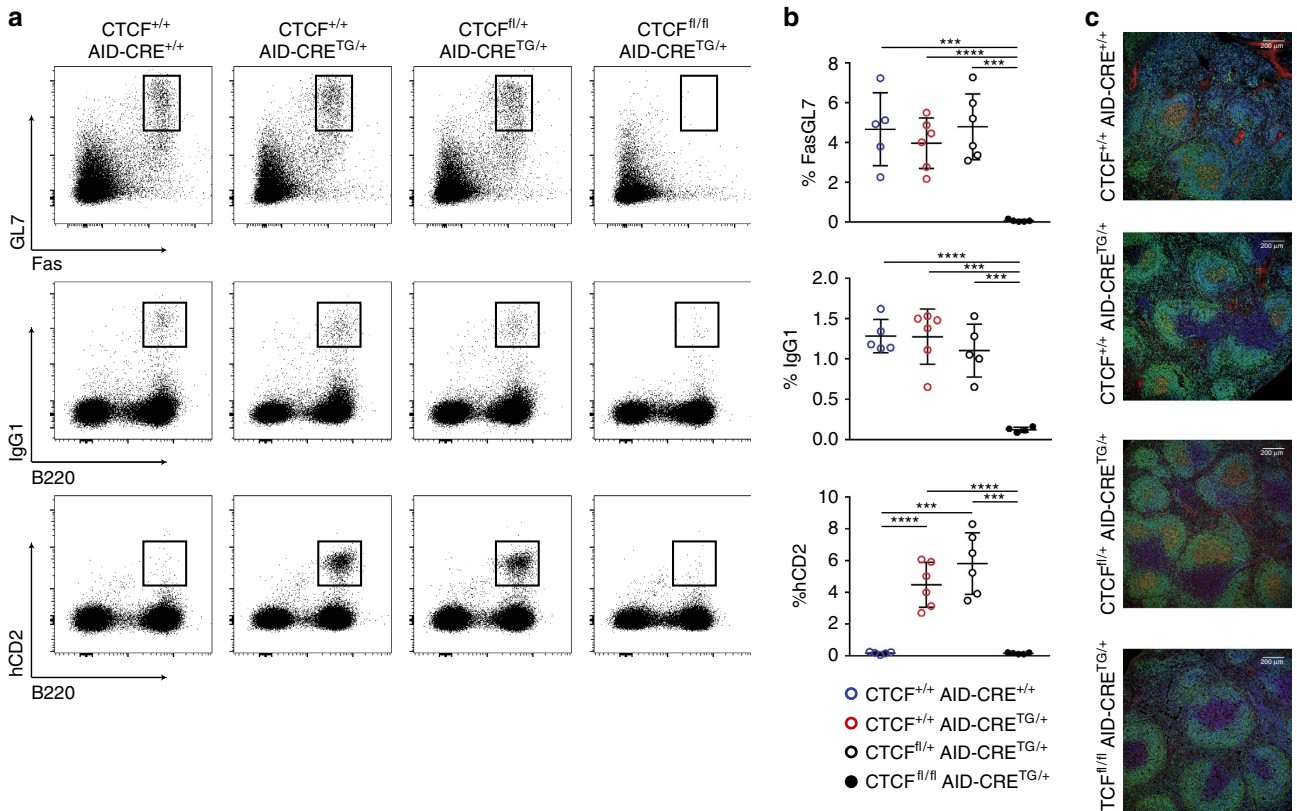

**Figure 1 | CTCF is required for the germinal centre reaction.** (**a**) FACS analysis of GL7, Fas, IgG1 and hCD2 expression in spleen B cells from $CTCF^{+/+}$; $AID\text{-}CRE^{+/+}$ ($n=5$), $CTCF^{+/+}$;$AID\text{-}CRE^{TG/+}$ ($n=6$), $CTCF^{fl/+}$;$AID\text{-}CRE^{TG/+}$ ($n=5$) and $CTCF^{fl/fl}$; $AID\text{-}CRE^{TG/+}$ ($n=5$) mice 7 days after SRBC immunization. Plots are gated on B220$^+$ cells (top) or on total live lymphocytes (middle and bottom). (**b**) Quantifications of GL7, Fas, IgG1 and hCD2 cells as percentages within B220$^+$ cells. ***$P<0.001$; ****$P<0.0001$. (**c**) Confocal immunofluorescence microscopy of immunized spleen cryosections from $CTCF^{+/+}$;$AID\text{-}CRE^{+/+}$, $CTCF^{+/+}$;$AID\text{-}CRE^{TG/+}$, $CTCF^{fl/+}$;$AID\text{-}CRE^{TG/+}$ and $CTCF^{fl/fl}$; $AID\text{-}CRE^{TG/+}$ mice. The sections were stained with anti-B220 (green), PNA (red) and DAPI. Scale bar, 200 μm. Mean values (**b**) ± s.d. are shown. Statistical analysis was done with two-tailed unpaired Student's *t*-test.

cells between CTCF deficient and control mice (82.84% ± 7.3% in CTCF$^{fl/+}$ cells and 84.31% ± 10.7% in CTCF$^{fl/fl}$ cells) (Fig. 2a,b). To rule out that the absence of phenotype could be due to a defective deletion of the CTCF-floxed allele, we measured the levels of CTCF in sorted hCD2$^+$ B cells from LPS/IL4 cultures (Fig. 2c). We observed a strong reduction of CTCF protein and mRNA levels in CTCF$^{fl/fl}$ compared with CTCF$^{fl/+}$ mice (Fig. 2d), suggesting that the disparity between the results obtained *in vivo* and *in vitro* is likely due to the failure of LPS/IL4 stimulation to faithfully recapitulate the need for CTCF *in vivo*.

To address this issue, we sought for *in vitro* stimulation conditions that could better mimic *in vivo* B-cell activation and decided to use a strategy where B cells can be stimulated *in vitro* by T cells in the presence of immobilized anti-CD3 together with soluble anti-CD28, as previously described[30–32]. Under these conditions, we observed that the population of hCD2 positive cells, used to track B-cell activation, was significantly reduced in the CTCF-deficient samples compared with control mice (72h: 17.34% ± 7.2% in CTCF$^{fl/+}$ cells versus 10.71% ± 2.9% in CTCF$^{fl/fl}$ cells; 96 h: 23.67% ± 7.2% in CTCF$^{fl/+}$ cells versus 15.33% ± 3.8% in CTCF$^{fl/fl}$ cells; percentages of hCD2$^+$ gated in B220$^+$ cells) (Fig. 2e,f). Quantification of CTCF protein in hCD2$^+$ cells revealed a similar degree of depletion to LPS/IL4 cultures (Fig. 2g), along with a significant reduction of CTCF mRNA levels (Fig. 2h). Therefore, the distinct impact of CTCF deletion in LPS/IL4 versus CD3/CD28 T-stimulated B cells was not due to differential degree of CTCF depletion as measured by protein levels (Fig. 2c,g and Supplementary Fig. 2) by RNA levels

(Fig. 2d,h) or allele excision (Supplementary Fig. 3a,b). Likewise, the increased sensitivity to CTCF loss of CD3/CD28-stimulated B cells could not be accounted for by an increased proliferation rate under these conditions (Supplementary Fig. 3c). Thus, our data suggest that similarly to GC B cells *in vivo*, B cells activated *in vitro* in the presence of T cells are sensitive to CTCF loss. In contrast, B cells stimulated by LPS and IL4 seem refractory to the absence of CTCF.

**T–B-cell co-cultures recapitulate the GC reaction.** To understand the differential requirement for CTCF of B cells under different activation cues, we first performed RNA-seq of hCD2$^+$ CTCF proficient B cells stimulated with LPS/IL4, hCD2$^+$ B cells stimulated with CD3/CD28 T cells and *in vivo* activated Fas$^+$ GL7$^+$ GC B cells (Supplementary Fig. 4a). Each of these samples was compared with RNA-seq data from resting splenic B cells and the three pairwise comparisons were first analysed with Venn diagrams (Fig. 3a,b). We observed that all three stimulation conditions shared a high proportion of both downregulated and upregulated genes. However, we found that CD3/CD28 T-stimulated B cells and GC B cells shared a specific set of 267 downregulated genes (shaded area 1), whereas LPS/IL4-stimulated B cells only shared 97 unique downregulated genes (shaded area 2) with GC B cells (Fig. 3a). Similar results were found for upregulated genes (Fig. 3b). Next, we selected the 10% most differentially expressed genes between GC and resting B cells, and compared their expression levels in all four conditions. Importantly,

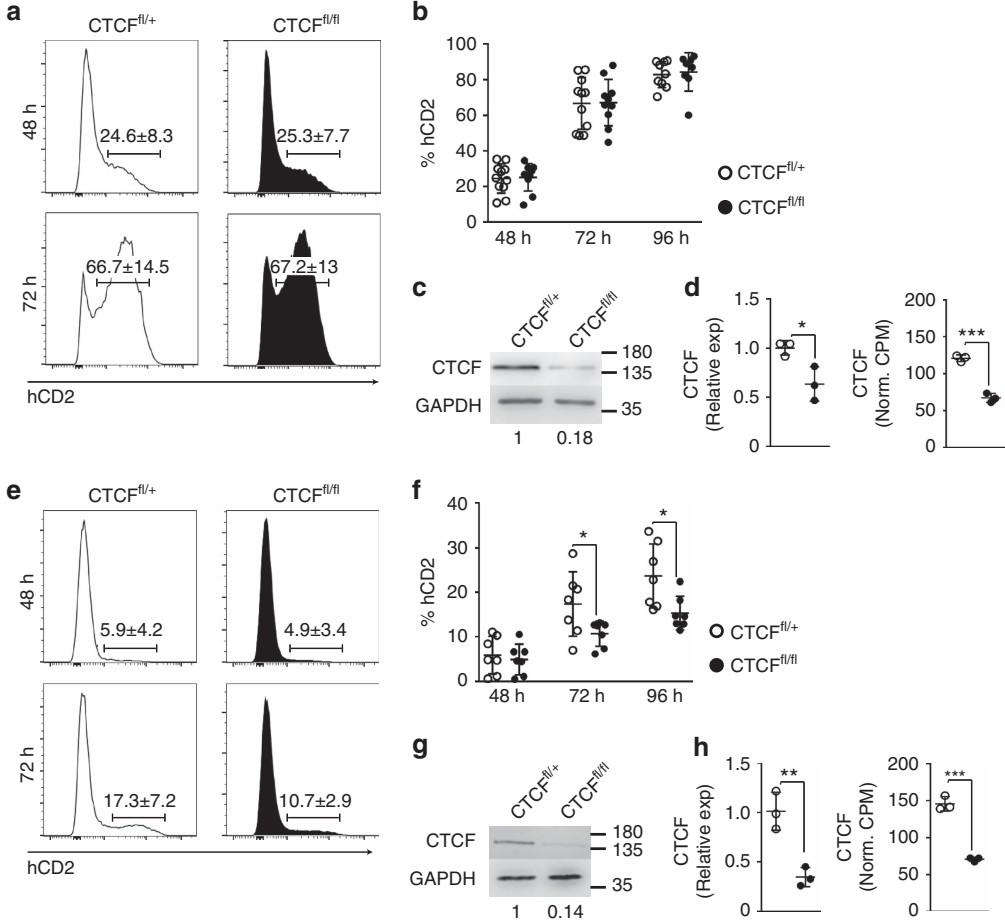

**Figure 2 | Activation context determines the requirement of activated B cells for CTCF.** (**a**) hCD2 expression in spleen B cells from CTCF$^{fl/+}$ and CTCF$^{fl/fl}$ mice after 48 and 72 h of LPS/IL4 stimulation. (**b**) Quantification of hCD2$^+$ cells from CTCF$^{fl/+}$ and CTCF$^{fl/fl}$ mice after 48, 72 and 96 h of LPS/IL4 stimulation. $n$(CTCF$^{fl/+}$) $=11$; $n$(CTCF$^{fl/fl}$) $=10$. (**c**) Western blot analysis of CTCF in isolated hCD2$^+$ B cells from CTCF$^{fl/+}$ and CTCF$^{fl/fl}$ mice after 72 h of LPS/IL4 stimulation. CTCF amount normalized to GAPDH is shown underneath. (**d**) CTCF mRNA quantification by qRT-PCR ($P = 0.0283$) and by RNA-seq ($P = 0.0003$). (**e**) hCD2 expression in spleen B cells from CTCF$^{fl/+}$ and CTCF$^{fl/fl}$ mice after 48 and 72 h of stimulation in CD3/CD28 T–B co-cultures. Histograms show hCD2 expression gated on B220 + B cells. (**f**) Quantification of hCD2$^+$ cells from CTCF$^{fl/+}$ and CTCF$^{fl/fl}$ mice after 48, 72 and 96 h of CD3/CD28 T-cell stimulation. Percentages of hCD2 + cells gated on B220 + B cells are shown. $n$(CTCF$^{fl/+}$) $=7$; $n$(CTCF$^{fl/fl}$) $=7$. $P$(72 h) $= 0.0438$; $P$(96 h) $= 0.0187$. (**g**) Western blot analysis of CTCF in isolated hCD2$^+$ B cells from CTCF$^{fl/+}$ and CTCF$^{fl/fl}$ mice after 72 h of stimulation with CD3/CD28 and T cells. CTCF amount normalized to GAPDH is shown underneath. (**h**) mRNA CTCF quantification by qRT-PCR ($P = 0.0059$) and by RNA-seq ($P = 0.0002$). Mean values (a and e) ± s.d. are shown. CTCF$^{fl/+}$, white dots; CTCF$^{fl/fl}$; black dots. Statistical analysis was done with two-tailed unpaired Student's $t$-test.

we found a remarkable similarity between GC and CD3/CD28 T-stimulated B cells in the expression of genes related with cell cycle and other key features of the GC reaction (Fig. 3c,d for representative examples of some functionally relevant genes in GC B cells, and Supplementary Fig. 4b). Together, these data indicate that co-culturing B cells with CD3/CD28-activated T cells recapitulates the *in vivo* GC transcriptional programme better than LPS/IL4 stimulation does, which in turn could explain the differential requirement for CTCF of B cells stimulated under these conditions.

**CTCF transcriptionally regulates key processes of GC biology.** To further characterize the role of CTCF in the GC reaction, and given the unfeasibility to obtain CTCF-deficient GC cells *in vivo*, we performed RNA-seq analysis of hCD2$^+$ B cells after *in vitro* stimulation with CD3/CD28 T cells or with LPS/IL4. We found that 2,229 genes were differentially expressed when we compared CTCF$^{fl/fl}$ and CTCF$^{fl/+}$ cells from CD3/CD28 T cultures (adjusted $P < 0.05$), approximately half of which were upregulated

(1,154) and half were downregulated (1,064) (Fig. 4a and Supplementary Data 1). In sharp contrast, only 50 genes were differentially expressed in CTCF$^{fl/fl}$ versus CTCF$^{fl/+}$ hCD2$^+$ cells from LPS/IL4 cultures (Fig. 4a and Supplementary Data 2). This notion was reinforced after plotting the $Z$-scores of the 20% most differentially expressed genes between CTCF-deficient and control CD3/CD28 T-stimulated B cells, which showed identical profiles in CTCF$^{fl/fl}$ and CTCF$^{fl/+}$ cells after LPS/IL4 stimulation (Fig. 4b). Together, these data agree with the finding that CTCF is dispensable for LPS/IL4-stimulated B cells but required for CD3/CD28 T-stimulated B cells and for GC B cells.

To gain mechanistic insights into the function of CTCF in GC B cells we first compared the transcriptional shifts induced in GC B cells (GC versus naive B cells) with those induced upon CTCF depletion (CTCF$^{fl/fl}$ versus CTCF$^{fl/+}$, CD3/CD28 T-stimulated B cells), and found that most (75.7%, 1,668 genes) of all genes controlled by CTCF are part of the GC transcriptional programme *in vivo* (Fig. 4c). Importantly, we observed that almost 65% (30.7% + 34.1%) of these common genes (1,095 out

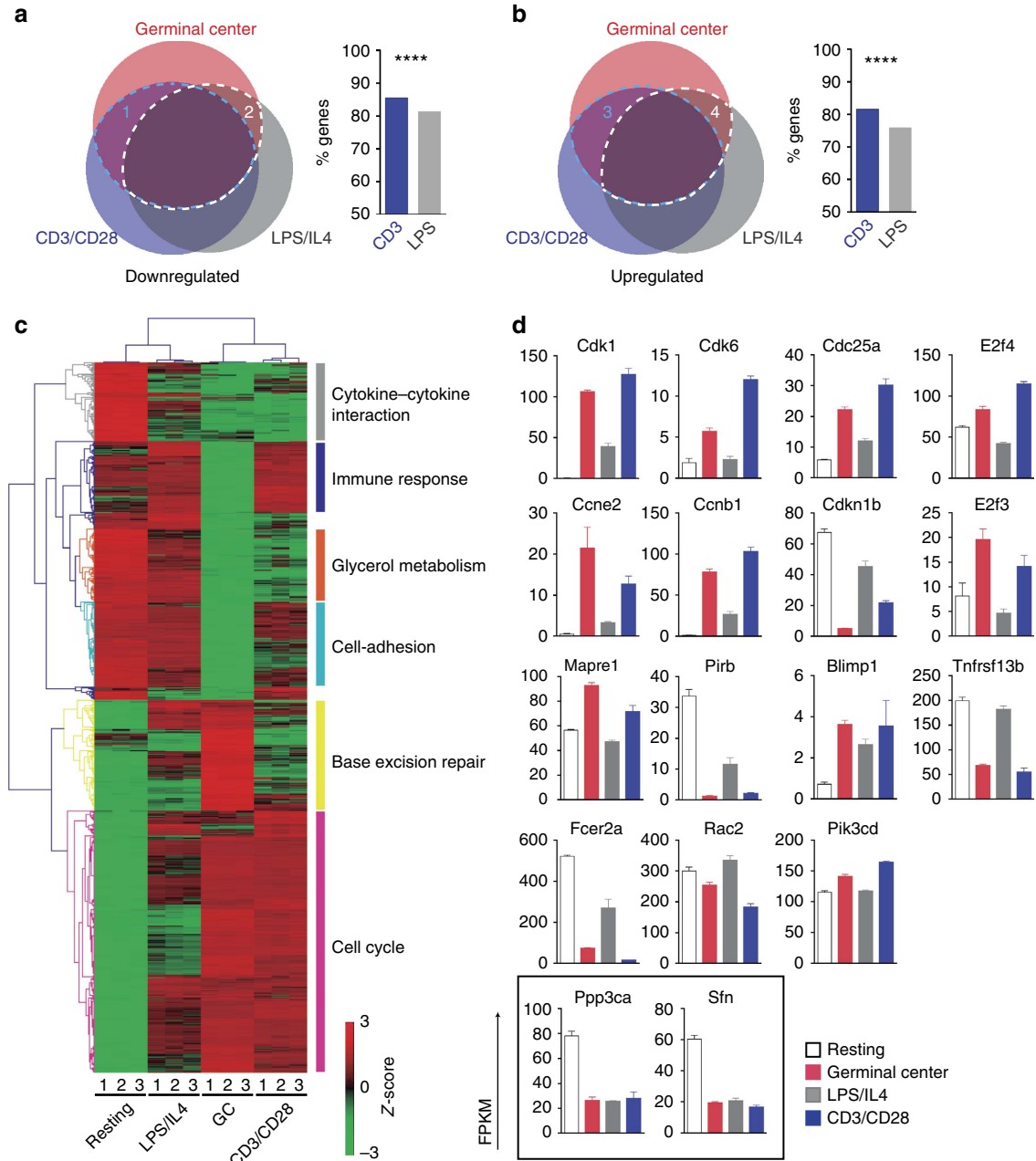

**Figure 3 | *In vitro* B cell stimulation through T cells recapitulates the GC reaction.** RNA-seq analysis was performed in triplicate samples from Fas+GL7+ GC B cells from Peyer's Patches, from sorted hCD2+ B cells from control mice stimulated for 48 h in CD3/CD28 T–B co-cultures or from hCD2+ B cells from wild-type mice stimulated for 48 h with LPS/IL4. Pairwise comparison of each data set with RNA-seq data from resting B cells is depicted as GC, CD3/CD28 and LPS/IL4, respectively. (**a,b**) Pairwise comparison of each data set with RNA-seq data from resting (R) B cells are depicted as GCvsR, CD3vsR and LPSvsR, respectively. Overlaps of downregulated (**a**) or upregulated (**b**) genes in all three conditions are depicted as Venn diagrams. Percentages of the common genes between GC and CD3/CD28 samples (blue dotted areas) or between GC and LPS/IL4 (white-dotted areas) are shown as bar graphs on the right. Statistical analyses were done with two-tailed $\chi^2$ analysis. $P < 0.00001$. (**c**) Heatmap analysis of the 10% most differentially expressed genes between resting and GC B cells. Z-scores for resting, GC, LPS/IL4 and CD3/D28-stimulated B cells are represented. Clustering was performed using the average linkage method based on Pearson correlation distance. Each column represents an independent replicate. Labels on the right indicate pathways found to be enriched by Gene Ontology Enrichment Analysis. (**d**) Expression profile of representative individual genes selected from **c**. For comparison, two genes with similar expression changes in GC, LPS/IL4 and CD3/CD28 sets are represented in the square below.

of 1,688) showed anti-correlative expression changes, that is, genes normally upregulated in GC cells are downregulated upon CTCF depletion, and vice versa (Fig. 4d). Pathway enrichment analysis of the genes differentially expressed upon CTCF depletion showed enrichment in terms included in the p53 pathway, B-cell receptor signalling, hematopoietic cell lineage and

cell cycle (Fig. 4e,f), among others. Analysis of individual genes revealed that CTCF depletion deregulates key players of cell-cycle progression and B-cell function (representative examples shown in Fig. 4f). The vast majority of these genes showed antagonistic shifts in expression levels when comparing GC and CTCF depletion programs (Fig. 4f). Together, our results indicate that

CTCF is critical to regulate the expression of key components of the GC transcriptional programme.

**CTCF deficiency recapitulates key features of PCs.** Our transcriptome results indicate that numerous genes involved in cell-cycle progression fail to be upregulated in CTCF-deficient cells (Fig. 4f). To address whether cell-cycle progression was affected by the lack of CTCF we performed Hoechst staining of CD3/CD28 T-stimulated CTCF$^{fl/fl}$ and CTCF$^{fl/+}$ B cells. We observed a reduced proportion of S/G2 phase CTCF$^{fl/fl}$ cells (42.8% ± 5.4%) when compared with control cells (52.9% ± 4.8%),

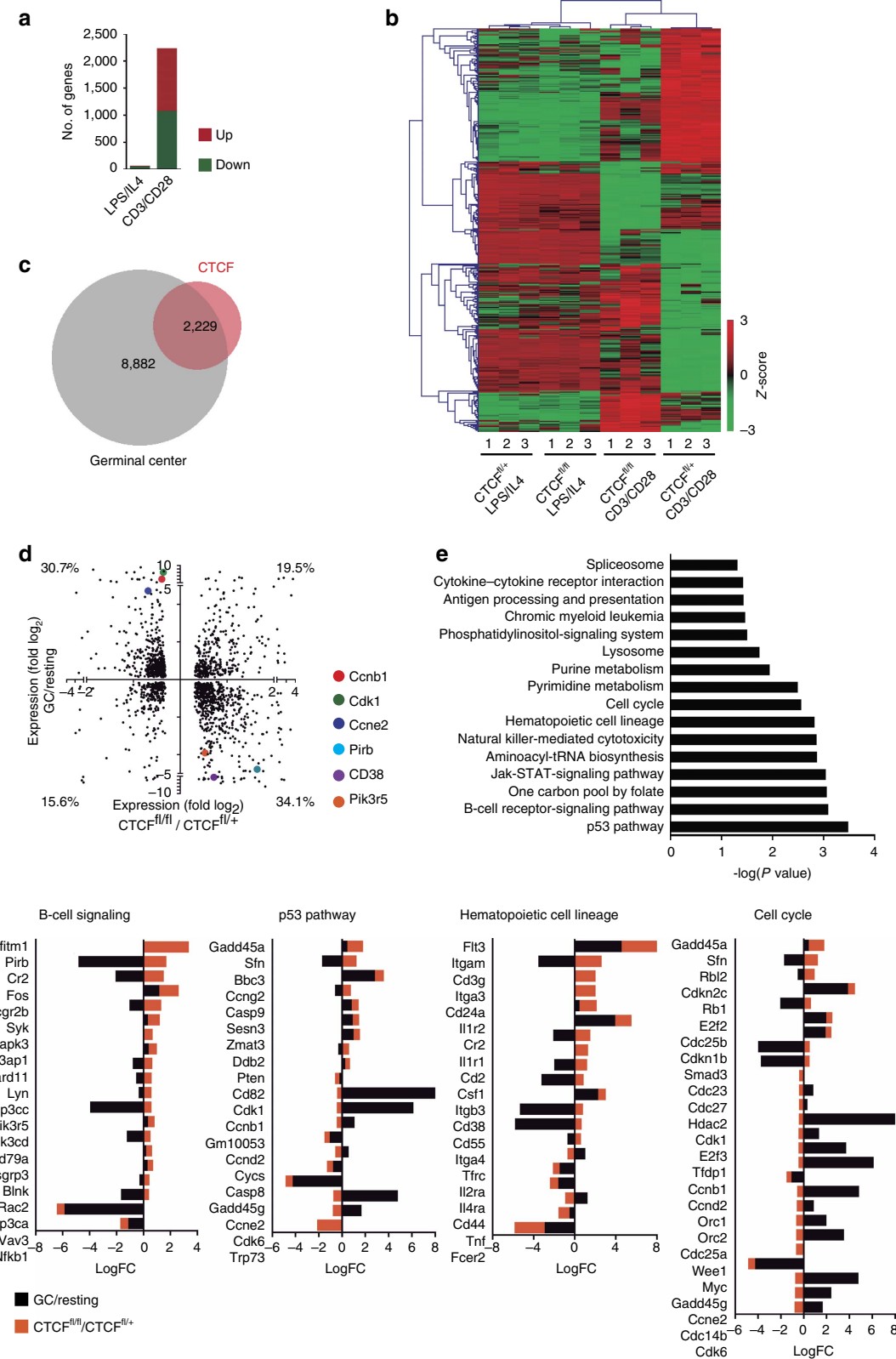

as well as an increase in the proportion of G1 phase in CTCF-deficient cells ($55.2\% \pm 5.6\%$ in CTCF$^{fl/fl}$ cells versus $44.7\% \pm 5.3\%$ in CTCF$^{fl/+}$ cells) (Fig. 5a). Expectedly, no proliferation defect was detected in LPS/IL4-stimulated B cells (Supplementary Fig. 5). Thus, CTCF deficiency impairs the proliferation of CD3/CD28 T-stimulated B cells.

GC and PC transcriptional programs are generally considered antagonistic processes at least in part due to the counter-regulation of the Bcl-6 and Blimp-1 master regulators. One of the hallmarks of GC to PC differentiation is a halt in cell proliferation. To address whether the proliferation defect found in CTCF-deficient B cells could be associated with transcriptional features of PC differentiation we compared the shifts found in the PC transcriptional programme (PC versus GC transcriptomes, Y axis)[33] with the transcriptional shifts induced by CTCF deficiency (CTCF$^{fl/fl}$ versus CTCF$^{fl/+}$ CD3/CD28-activated B cells, X axis). In contrast to the anti-correlative shifts found when compared to the GC transition (Fig. 4d) we found a positive correlation between the transcriptional shifts at the transition from GC to PC and those promoted by CTCF depletion. Specifically, about 60% of the genes showed the same sign of change in both conditions, that is, were either upregulated both at the GC to PC transition and upon CTCF depletion (26.4%) or downregulated in both cases (34%) (Fig. 5b). A major category of genes showing parallel regulation in the PC transition and upon CTCF deletion was cell-cycle genes (Fig. 5c). These data agrees with the observation that CTCF expression decreases at the transition from GC to PC differentiation (Fig. 5d, data from GSE60927 (ref. 33)). Remarkably, one of the genes upregulated upon CTCF depletion was the master regulator of PC differentiation, Blimp-1 (refs 7,8) (Fig. 5e). Accordingly, we found reciprocal downregulation of Bcl-6 upon CTCF depletion (Fig. 5e). To address the mechanism by which CTCF controls Blimp-1, we first analysed published data[34] and found that CTCF binds to Blimp-1 (Prdm1 gene) in B cells (Fig. 5f). Further, we performed chromatin immunoprecipitation (ChIP)-quantitative PCR (qPCR) in CD3/CD28 T-cell-stimulated B cells, and confirmed that under these experimental conditions CTCF also binds to Blimp-1 (Fig. 5e), suggesting that direct binding of CTCF to Prdm1 gene could mediate a chromatin configuration that hinders Blimp-1 expression. In addition, numerous PC genes were upregulated, including immunoglobulin genes or the PC marker Syndecan-1 and genes considered direct targets of Blimp-1, such as Myc, CIITA, Bcl11a or Lta were downregulated (Fig. 5h,i). Finally, we detected an increase in the secretion of IgM in CD3/CD28 T-activated CTCF$^{fl/fl}$ B cells when compared with CTCF$^{fl/+}$ controls (Fig. 5j), suggesting that the generation of antibody secreting cells is more efficient in the absence of CTCF. Together these results support the notion that CTCF maintains the GC programme by preventing the premature activation of the PC programme.

**CD40 signalling partially rescues CTCF deficiency.** To address the contribution of Blimp-1 to the phenotype observed in CTCF-deficient B cells we cultured B cells in the presence of T cells and CD3/CD28 and then added anti-CD40 (Fig. 6a), since signalling through the CD40 receptor has been shown to downregulate Blimp-1 levels in B cells[35–37]. qRT-PCR analysis confirmed once again that CTCF-deficient B cells have higher Blimp-1 expression levels than their littermate controls (Fig. 6b, $0.92\% \pm 0.17\%$ in CTCF$^{fl/+}$ versus $2.97\% \pm 0.7\%$ in CTCF$^{fl/fl}$). Importantly, this effect was dampened in cells treated with anti-CD40 (Fig. 6b) ($2.97\% \pm 0.7\%$ in CTCF$^{fl/fl}$ − anti-CD40 cells versus $1.60\% \pm 0.6\%$ in CTCF$^{fl/fl}$ + anti-CD40 cells).

Next, we determined the proportion of B cells in the CD3/CD28 T-cell cultures after 48h of treatment with anti-CD40 (Fig. 6c). We observed that in the absence of anti-CD40, the proportion of B220$^+$ B cells in the CTCF$^{fl/fl}$ B cell co-cultures is reduced by 53% when compared with CTCF$^{fl/+}$ B cell co-cultures, again reinforcing the idea that CTCF is important for B-cell proliferation (Fig. 6c, left) ($42.2\% \pm 8.5\%$ in CTCF$^{fl/+}$ cells versus $22.4\% \pm 6.9\%$ in CTCF$^{fl/fl}$ cells). However, in the presence of anti-CD40, this reduction is attenuated to 77% ($37.7\% \pm 10\%$ in CTCF$^{fl/fl}$ cells versus $49\% \pm 14.1\%$ in CTCF$^{fl/+}$ control cells; $P = 0.2$) (Fig. 6c, right). Cell-cycle analysis expectedly revealed that CTCF$^{fl/fl}$ cultures contained a lower proportion of cycling B cells than CTCF$^{fl/+}$ B cells ($35.7\% \pm 7.3\%$ versus $21\% \pm 4.8\%$) (Fig. 6d, left). In contrast, after treatment with anti-CD40, the proportion of S/G2 cells was reduced to a much milder extent ($28.5$ versus $37\%$, $P > 0.05$) (Fig. 6d, right). These results indicate that signalling through the CD40 receptor and downregulation of Blimp-1 partially rescue the cell-cycle impairment associated with CTCF deficiency.

## Discussion

The establishment of the GC reaction and the shift to PC differentiation involve complex transcriptional programs and the coordinated expression of gene networks. While the function of master transcriptional regulators in these events, such as Bcl-6 and Blimp-1 is firmly established, the impact of chromatin structure changes at the GC to PC transition is poorly understood. CTCF is believed to play general transcriptional regulatory functions by establishing long-range DNA interactions between distal enhancers and promoters (reviewed in ref. 20). We have shown here that CTCF is absolutely required for the GC reaction through the transcriptional regulation of genes required for B cell proliferation, and that this effect is at least partially due to the inhibition of Blimp-1, which programs GC B cells for their entry into PC differentiation.

We report here that the sensitivity of in vitro activated B cells to CTCF loss is highly dependent on the activation pathway. This is an unexpected finding, given that all three scenarios of

**Figure 4 | CTCF transcriptionally regulates B cell signalling and cell cycle.** RNA-seq analysis was performed in triplicates from hCD2$^+$ sorted B cells from CTCF$^{fl/fl}$ and CTCF$^{fl/+}$ mice after 48 h stimulation in the presence of CD3/CD28 T or LPS/IL4. Differentially expressed genes (adjusted $P < 0.05$) were subject to further analysis. (**a**) Number of differentially transcribed genes between CTCF$^{fl/fl}$ and CTCF$^{fl/+}$ B cells stimulated LPS/IL4 (50 differentially expressed genes) or CD3/CD28 stimulation (2,229 differentially expressed genes) conditions. Red: upregulated genes; green: downregulated genes. Adjusted $P < 0.05$. See Supplementary Data 1 and Supplementary Data 2 for complete list of genes. (**b**) Heatmap analysis of the 20% most differentially expressed genes between CTCF$^{fl/fl}$ and CTCF$^{fl/+}$ B cells after CD3/CD28 T stimulation. Z-scores values are represented. Clustering was performed using the average linkage method based on Pearson correlation distance. Each column represents an independent replicate. (**c**) Overlap between genes differentially expressed in GC reaction (wild-type GC B cells versus wild-type-resting B cells, grey circle) and genes differentially expressed upon CTCF depletion in CD3/CD28 T-stimulated B cells (CTCF$^{fl/fl}$ versus CTCF$^{fl/+}$ hCD2$^+$ sorted B cells, red circle). (**d**) Two-dimension expression plot. X axis, log2 fold values of significantly changed genes upon CTCF deficiency (CTCF$^{fl/fl}$ versus CTCF$^{fl/+}$, CD3/CD28-stimulated B cells). Y axis, log2 fold values of significantly changed genes in GC versus resting B cells. Percentage of genes in each quadrant is shown. (**e**) KEGG pathway enrichment analysis of genes differentially expressed in CTCF$^{fl/fl}$ versus CTCF$^{fl/+}$ CD3/CD28-stimulated B cells. (**f**) Log fold change (LogFC) representation of the genes included in the KEGG pathways analysed in (**e**). Black, GC versus resting B cells fold change. Orange, CTCF$^{fl/fl}$ versus CTCF$^{fl/+}$ CD3/CD28 T-stimulated splenic B cells fold change.

B cell activation analysed here (that is, *in vivo* GC B cells, LPS/IL4 activation or CD3/CD28 T-dependent activation) share a big proportion of transcriptome changes, and that all three involve an intense proliferation rate. However, our data are reminiscent of the finding that a proliferation defect in CTCF-deficient T cells can be bypassed with phorbol ester and ionomycin stimulation,

which circumvents TCR signalling[38]. Here we show that LPS/IL4-stimulated B cells seem refractory to the loss of CTCF whereas *in vivo* and in *in vitro* T co-cultures, the B-cell activation programme is clearly impaired. This finding is supported by the higher resemblance between GC and T-co-cultures transcriptomes. Therefore our data show that LPS/IL4 activation, one of

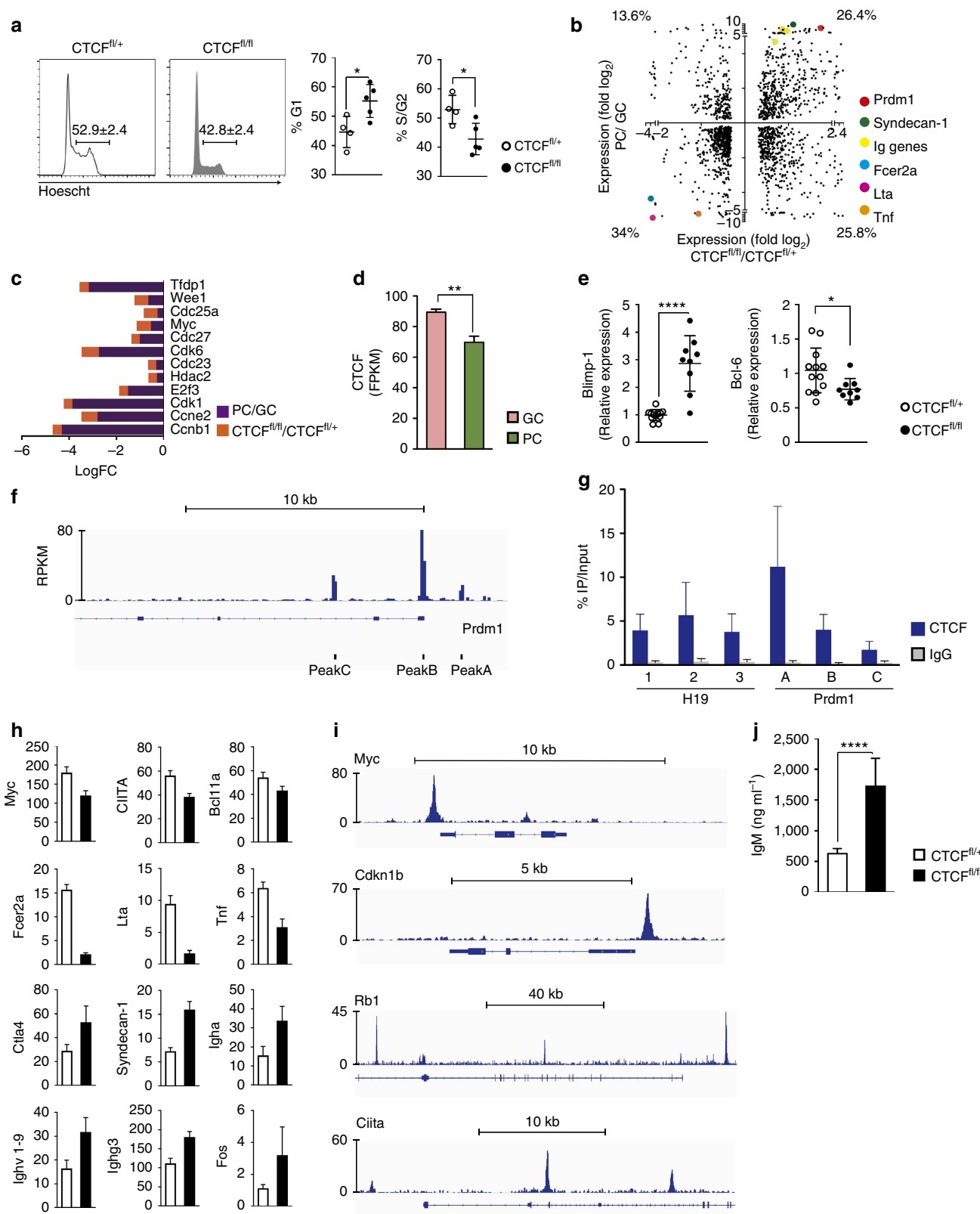

the most frequently used protocols for B cell activation *in vitro*, falls short of recapitulating features of the *in vivo* GC reaction as compared to *in vitro* T-cell-mediated activation. Although beyond the scope of this work, further molecular characterization of these activation programs is a fascinating topic for further investigation.

The insensitivity of LPS/IL4-stimulated B cells to CTCF loss was molecularly supported by the finding that only 50 genes showed altered transcriptional levels in CTCF-deficient B cells, as opposed to >2,000 genes significantly altered in B cells stimulated by T cells. Of note, the amount of remaining CTCF protein was comparable under both conditions and roughly 15% of normal levels (within $CD2^+$ sorted cells). Although we cannot discriminate whether this reflects a small proportion of cells that retain unrecombined floxed alleles or a small proportion of CTCF protein in many cells due to slow protein turnover, our data clearly show that T-stimulated B cells sense this suboptimal presence of CTCF, whereas LPS/IL4-stimulated cells do not. We speculate that the strong signalling triggered by cytokines bypasses the need for CTCF whereas in the context of T-cell-mediated activation *in vitro* or *in vivo*, where the activation thresholds may become limiting, CTCF turns into a critical transcriptional regulator. In this regard, residual amounts of CTCF in LPS/IL4 cells might be enough to provide minimal architectural requirements for B-cell proliferation and survival in the presence of cytokines. In sharp contrast, the complete absence of GC B cells in immunized CTCF-deficient mice, suggests that *in vivo* the absence of CTCF promotes either a more profound block in proliferation or additional functional defects. This is an important issue that deserves further analysis with suitable mouse models.

While numerous studies have approached the DNA-binding profile of CTCF in different cell types and tissues, the function of CTCF in global transcriptional regulation has only rarely been addressed. Strikingly, ChIP-seq analyses show that about 84% of all CTCF-binding sites found in LPS/IL4-stimulated B cells are shared with naive B cells (Supplementary Fig. 6) our unpublished analysis from ref. 39. In addition, establishing a functional correlation between a CTCF DNA-binding site and its targets for transcriptional regulation is remarkably complex[40]. Thus, our approach has directly focused on the functional impact of CTCF deficiency by measuring the transcriptional shifts in GC B cells and showed that CTCF is required for the transcriptional regulation of more than 2,000 genes either directly or indirectly.

Importantly, the vast majority of CTCF-regulated genes are part of the GC transcriptome, indicating that CTCF plays a pivotal function in the initiation and/or maintenance of the GC transcriptional programme. While it is likely that CTCF is required for GC initiation, the delay of CTCF depletion—dependent on AID expression—in our model is not suitable to specifically ascertain that point. In contrast, our data strongly support the view that CTCF is required for the maintenance of the GC transcriptional programme and that it does so by repressing the transition to PC differentiation. First, CTCF deletion promotes a delay in cell proliferation, one of the hallmarks of B cells exiting the GC programme. Second, we found abnormally high levels of Blimp-1, the master regulator of PC differentiation, in CTCF-deficient cells. Given that we only find a mild decrease in Bcl-6 levels under these conditions, we believe that this effect is secondary to Blimp-1 upregulation, and not vice versa. Third, numerous other genes of the PC differentiation programme are upregulated upon CTCF deficiency. Fourth, we detected increased levels of IgM secretion in CTCF-deficient cultures. Finally, restoring Blimp-1 levels by anti-CD40 treatment partially rescues the proliferation rate of CTCF-deficient cells, again suggesting that the proliferation defect is secondary to the premature trigger of PC differentiation. Thus, our data reveal a key function of CTCF in orchestrating transcriptional changes required for the GC programme and for preventing premature PC differentiation through Blimp-1 inhibition.

## Methods

**Mice and immunizations.** Conditional CTCF-deficient mice were obtained by breeding $CTCF^{fl/fl}$ mice[41] with $AID-CRE^{+/TG}$(ref. 42). For experiments shown in Fig. 1 $CTCF^{fl/fl}$; $AID-CRE^{TG/+}$, $CTCF^{fl/+}$; $AID-CRE^{TG/+}$, $CTCF^{+/+}$; $AID-CRE^{TG/+}$ and $CTCF^{+/+}$; $AID-CRE^{+/+}$ mice were obtained after breeding $CTCF^{fl/+}$; $AID-CRE^{TG/+}$ to $CTCF^{fl/+}$ mice. For the rest of the experiments in this work, $CTCF^{fl/fl}$; $AID-CRE^{TG/+}$, $CTCF^{fl/+}$; $AID-CRE^{TG/+}$ littermates were used, obtained by $CTCF^{fl/+}$ $AID-CRE^{TG/+}$ to $CTCF^{fl/fl}$ breeding. All animals had previously bred to C57/BL6 background. All animal procedures were conducted in accordance with EU Directive 2010/63/UE, enforced in Spanish law under Real Decreto 53/2013. The procedures have been reviewed by the Institutional Animal Care and Use Committee (IACUC) of Centro Nacional de Investigaciones Cardiovasculares, and approved by Consejeria de Medio Ambiente, Administración Local y Ordenación del Territorio of Comunidad de Madrid (Ref: PROEX 341/14). T-dependent immunizations were performed by intravenous injection of $10^8$ SRBCs resuspended in 100 μl of sterile PBS. Immunization response was analysed in spleen 7 days after injection. Mice were housed in specific pathogen-free conditions. Male and female mice between 7–13 weeks were used for the experiments. Number of animals per group to detect biologically significant effect sizes was calculated using appropriate statistical sample size formula and indicated in the biometrical planning section of the animal license submitted to the Consejeria de Medio Ambiente, Administración Local y Ordenación del Territorio of Comunidad de Madrid (Ref: PROEX 341/14). Blinding and randomization was not applicable to the animal studies.

**Cell cultures.** For LPS/IL4 cultures, primary B cells were purified by immuno-magnetic depletion using anti-CD43 beads (Miltenyi Biotec) and cultured at a final concentration of $1.2 \times 10^6$ cells per ml in complete RPMI supplemented with 10% FBS, 50 μM of 2-βMercaptoethanol (Gibco), 20 mM Hepes (Gibco), 10 ng ml$^{-1}$ of IL4 (PeproTech) and 25 μg ml$^{-1}$ lipopolysaccharide (LPS, Sigma-Aldrich). In the T–B-cell co-culture, $CD43^+$ cells were added at a ratio 1:1 to $CD43^-$ B cells in the presence of plastic-bound anti-CD3 (Tonbo, 5 μg ml$^{-1}$) and soluble anti-CD28 (BioXcell, 2 μg ml$^{-1}$) for T-cell stimulation. When indicated, cells were treated with anti-mouse CD40 (BD Pharmigen, 1 μg ml$^{-1}$).

---

**Figure 5 | CTCF deficiency recapitulates key features of plasma cells.** (**a**) FACS cell-cycle analysis of $hCD2^+$ cells from $CTCF^{fl/+}$ (*n* = 4) and $CTCF^{fl/fl}$ (*n* = 5) mice after 48 h of stimulation with CD3/CD28 and T cells. Numbers indicate percentages ± s.d. Quantification of G1 and S/G2 phase proportions is shown on the right. p(G1) = 0.0242; p(S/G2) = 0.0221, two-tailed Student's *t*-test. (**b**) Two-dimension expression plot. *X* axis, log2 fold values of significantly changed genes upon CTCF deficiency ($CTCF^{fl/fl}$ versus $CTCF^{fl/+}$, CD3/CD28-stimulated B cells). *Y* axis, log2 fold values of significantly changed genes in PC versus GC B cells (data extracted from ref. 33. Coloured dots highlight genes of the PC differentiation programme. (**c**) Quantification of significantly changed cell-cycle-related genes. Log fold change (LogFC) in PC versus GC B cells (purple) and in $CTCF^{fl/fl}$ versus $CTCF^{fl/+}$ CD3/CD28 T-stimulated B cells (orange). (**d**) RNA-seq analysis of CTCF expression in GC B cells and PC (data extracted from ref. 33). (**e**) qRT-PCR analysis of Blimp-1 (*Prdm1*) and Bcl-6 expression in CD3/CD28 T-stimulated $CTCF^{fl/+}$ (*n* = 12) and $CTCF^{fl/fl}$ (*n* = 9) $hCD2^+$ cells. *P*(Blimp-1) < 0.0001; *P*(Bcl-6) = 0.0331. (**f**) CTCF binding at the Prdm1 locus in spleen B cells stimulated with LPS/IL4 (data from ref. 34). (**g**) CTCF ChIP-qPCR on Peaks A–C of the Prdm1 (Blimp-1) locus in spleen B cells stimulated with CD3/CD28 and T cells for 48 h (*n* = 3). Data show per cent of input. IgG ChIP is shown as negative control. H19 Peaks 1–3 are shown as positive control for CTCF ChIP. Error bars indicate s.d. (**h**) RNA-seq analysis of Blimp-1 (Prmd1) targets in $hCD2^+$ cells from CTCFfl/+ (*n* = 3) and CTCFfl/fl (*n* = 3) mice stimulated with CD3/CD28 and T cells for 48 h. (**i**) Blimp-1 binding at *Myc*, *Cdkn1b*, *Rb1* and *Ciita* locus in plasmablasts (data from ref. 9) (**j**) ELISA quantification of IgM secretion from CD3/CD28 T-stimulated $CTCF^{fl/+}$ (*n* = 3) and $CTCF^{fl/fl}$ (*n* = 3) cells for 72 h (*P* = 0.0151). ($CTCF^{fl/+}$, white; $CTCF^{fl/fl}$; black). Statistical analysis was done with two-tailed unpaired Student's *t*-test.

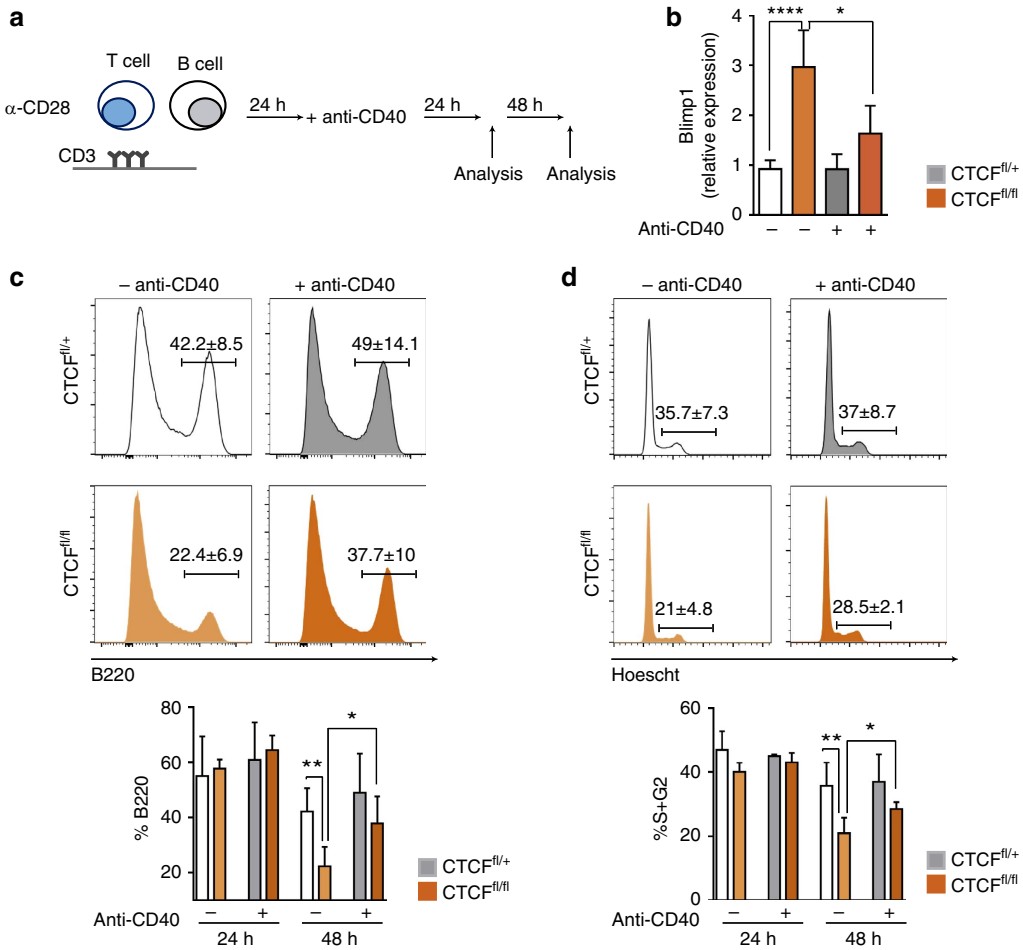

**Figure 6 | CD40 signalling rescues Blimp-1 expression levels and cell proliferation in CTCF-deficient cells.** CTCF^fl/+ (n = 4) and CTCF^fl/fl (n = 5) splenic B cells stimulated with CD3/CD28 T cells were treated with anti-CD40 antibody (+ anti-CD40), and compared to cells left untreated (− anti-CD40). (**a**) Representation of the experimental approach. (**b**) Analysis of Blimp-1 (*Prdm1*) mRNA by qRT-PCR in hCD2+ CTCF^fl/+ or CTCF^fl/fl cells after 48 h without or with anti-CD40 treatment. ***P = 0.0001, *P = 0.0159. (**c**) Histograms show the percentage of B220+ B cells in CTCF^fl/+ and CTCF^fl/fl B-T CD3/CD28 co-cultures after 48 h of anti-CD40 treatment. Lower bar graph shows quantification of B220 proportion at 24 and 48 h after CD40 treatment. **P = 0.0063, *P = 0.023. (**d**) Histograms show the percentage of cycling (S + G2/M) cells in CTCF^fl/+ and CTCF^fl/fl B-T CD3/CD28 co-cultures after 48 h of anti-CD40 treatment. Lower bar graph shows quantification of cycling cells at 24 and 48 h after CD40 treatment. **P = 0.0083, *P = 0.0126. Mean values ± s.d. are shown. Statistical analysis was done with two-tailed unpaired Student's *t*-test.

**Flow cytometry.** Single-cell suspensions were obtained from spleen after erythrocyte lysis, and stained with fluorophore-conjugated anti-mouse or human antibodies (BD Pharmigen or Invitrogen) to detect B220 (RA3-6B2, 1/200), Fas (Jo2, 1/400), GL7 (1/200), hCD2 (S5.5, 1/200), immunoglobulin A (IgA) (1/200), immunoglobulin G1 (IgG1, A85-1, 1/400), CD19 (ID3, 1/400), CD25 (1/100) immunoglobulin M (IgM) (1/200), immunoglobulin D (IgD, 11-26, 1/200), CD21 (7G6, 1/200), CD23 (B3B4, 1/200) and CD93 (AA4.1, 1/200). Cell-cycle analysis was performed with Hoechst33342 staining of alive B cells for 60 min. Samples were acquired on LSRFortessa or FACSCanto instruments (BD Biosciences) and analysed with FlowJo software. For preparative flow cytometry FACSAria (BD Biosciences) or SY3200 (Sony) sorters were used.

**RNA sequencing.** For RNA-seq analysis hCD2+ B cells were sorted from pooled CTCF^fl/fl or CTCF^fl/+ cultures after 48 h stimulation of CD3/CD28 or LPS/IL4 cultures (2–3 mice per sample) and RNA was isolated using the Qiagen RNeasy kit (Cat 74104). RNA from sorted Peyer's patch CD19+ Fas+ GL7+ cells and CD19+ Fas-GL7- cells were used as GC and resting B-cell controls, respectively. For RNA-seq library preparation, 500 ng of total RNA were processed using the TruSeq RNA sample preparation kit v2 (Illumina RS-122-2001) following manufacturer's instructions. In brief, poly A + RNA was purified using poly-T oligo-attached magnetic beads with two rounds of purification followed by fragmentation and first and second cDNA strand synthesis. Next, cDNA 3′ ends were adenylated and the adaptors were ligated followed by PCR library amplification. Finally, the size of the libraries was checked using the Agilent 2100 Bioanalyzer DNA 1000 chip and their concentration was determined using the Qubit fluorometer (Life Technologies). Libraries were sequenced on a HiSeq2500

(Illumina) to generate 60 × single reads. FastQ files for each sample were obtained using CASAVA v1.8 software (Illumina). RNA-seq experiments were performed in the Genomics Unit of the CNIC. Sequencing reads were pre-processed with a pipeline that included FastQC[43] and Cutadapt[44] filtering and trimming. The resulting reads were mapped using the mouse transcriptome (GRCm38, release 76; aug2014 archive) and quantified using RSEM v1.17 (ref. 45). Data were then processed with a differential expression analysis pipeline that used Bioconductor package EdgeR (ref. 46) for normalization and differential expression testing. Pathway enrichment analysis was done with the Ingenuity Pathways Analysis software (Qiagen) and with GO enrichment analysis tool (Gene Ontology Consortium).

**Immunofluorescence.** Spleens were fixed with paraformaldehyde 4% for 2 h at room temperature, incubated in 30% sucrose overnight at 4 °C, embedded in Tissue Tek O.C.T. (Sakura Finetek) and frozen at –80 °C. Sections of 10 μm thickness were prepared. The following antibodies were used: rat anti-B220 (Miltenyi, 1/100), chicken anti-rat Alexa Fluor 488 (Molecular Probes, 1/500), PNA-Alexa Fluor 647 (Life technologies, 1/100). Nuclei were counterstained with DAPI and slides were mounted with ProLong Gold (Life technologies). Images were acquired on a Leica SP5 confocal laser-scanning microscope.

**Immunoblotting.** hCD2+ B cells sorted from LPS/IL4 or CD3/CD28 cultures were incubated on ice for 20 min in RIPA lysis buffer in the presence of protease inhibitors (Roche) and lysates were cleared by centrifugation. Total protein was size-fractionated on SDS-PAGE 8% acrylamide–bisacrylamide gels and transferred to Protan nitrocellulose membrane (Whatman) in transfer buffer (0.19M glycine,

25 mM Tris base and 0.01% SDS) containing 20% methanol (90 min at 0.4A). Membranes were probed with anti-mouse-CTCF (1/2,500, Bethyl laboratories) and anti-mouse-tubulin (1/5,000, Sigma-Aldrich). Then, membranes were incubated with HRP-conjugated anti-rabbit (1/10,000, DAKO) and anti-mouse (1/10,000, DAKO) antibodies, respectively, and developed with Amesham ECL Western Blotting Detection Reagent (GE Healthcare Life Sciences). Quantification of band intensities was performed with ImageJ software.

**qPCR.** RNA was extracted from hCD2$^+$ B cells from CTCF$^{fl/fl}$ or CTCF$^{fl/+}$ mice after 48 h of stimulation with LPS/IL4 or CD3/CD28 and T cells using the Qiagen RNeasy kit and treated with DNAse. cDNA was synthesized using random hexamers (Roche) and SuperScript II reverse transcriptase. cDNA was quantified by SYBR green assay (Applied Biosystems) and normalized to GAPDH expression in triplicates. The following primers were used: mouse-GAPDH (forward) 5′-TGA AGC AGG CAT CTG AGG G-3′, (reverse) 5′-CGA AGG TGG AAG AGT GGG AG-3′; mouse-PRDM1 (forward) 5′-GCA AAG AGG TTA TTG GCG T-3′, (reverse) 5′-TGT AGA CTT CAC CGA TGA GG-3′; mouse-Bcl-6 (forward) 5′-ATG TAC CTG CAG ATG GAG CAT G-3′; mouse-Bcl-6 (reverse) 5′- ATC AGC ATC CGG CTG TTC A-3′; mouse-CTCF (forward) 5′- CAC CTG GGT CCT AAC AGA ACA GA-3′; mouse-CTCF (reverse) 5′- AGT ATG AGA GCG AAT GTG TCG TTT-3′.

Genomic DNA was isolated from hCD2+ B cells from CTCF$^{fl/fl}$ or CTCF$^{fl/+}$ mice after 48 h of stimulation with LPS/IL4 or CD3/CD28 and T cells. The following primers were used: CTCF deletion (forward) 5′- GGGCATCA-GATCTCATTAAGGA -3′; CTCF deletion (reverse) 5′- ACTCCATCTCTAGC-CTCTCTATT-3′.

**ChIP-qPCR.** ChIP was performed according to the Diagenode protocol (iDeal ChIP-seq Kit for Transcription Factors C01010055). In brief, cells were crosslinked in 1% formaldehyde (Sigma) for 10 min at 37 °C and sonicated with three rounds of 25 cycles 30 s ON/30 s OFF using a Bioruptor (Diagenode). Sonicated chromatin was incubated overnight at 4 °C either with anti-CTCF antibody (Diagenode) or an IgG control. DNA was quantified by qPCR using SYBR green (Applied Biosystems) in triplicates. The following primers were used: Prdm1-PeakA (forward) 5′-GGGGTTGTAGGTCCACCTGT-3′; Prdm1-PeakA (reverse) 5′- CTGGCACAAGAGCAAGCTAA -3′; Prdm1-PeakB (forward) 5′- ACTGGAGGGCCGAGTGTC -3′; Prdm1-PeakB (reverse) 5′- GGGAGGGG-GAAGAGTAGTCA -3′; Prdm1-PeakC (forward) 5′- GACACCAAGAGGGA-CCAGAG-3′; Prdm1-PeakC (reverse) 5′-AACTTCCCCGAAGGCTAGAG -3′; H19-Peak1 (forward) 5′- GTCACTCAGGCATAGCATTC-3′; H19-Peak1 (reverse) 5′- GTCTGCCGAGCAATATGTAG-3′; H19-Peak2 (forward) 5′-CAGTTGTGT-TTCTGGAGGG -3′; H19-Peak2 (reverse) 5′-TAGGAGTATGCTGCCACC -3′; H19-Peak3 (forward) 5′-TCTTTAGGTTTGGCGCAATCGA -3′; H19-Peak3 (reverse) 5′- GACGTCTGCTGAATCAGTTGTG-3′.

**ELISA.** Total IgM levels in supernatant cultures were measured using a Mouse ELISA Quantification Set (Bethyl Laboratories) according to manufacturer's instructions. In brief, supernatant from CTCF$^{fl/fl}$ and CTCF$^{fl/+}$ cultured in presence of T cells (CD3/CD28 T-stimulated cells) was collected after 72 h of stimulation. ELISA plates were coated with goat anti-mouse IgM (1/100, Bethy) and total IgM in supernatants was detected with HRP anti-mouse IgM (1/150,000, Bethyl). Optical densities were determined at 405 nm with a Benchmark Plus Microplate Reader (Bio-Rad). Absorbance from culture medium (culture medium without cells was used as negative control) was subtracted and concentrations were calculated by the interpolation of calibration curves.

**Statistics.** Statistical analyses were performed with GraphPad Prism (version 6.01 for Windows, GraphPad Software, San Diego, CA, USA) using two-tailed Student's $t$-test for all parameters conforming to normal distributions (according to Shapiro–Wilk normality test). Variance similarity was assessed with $F$ test. $\chi^2$ test was used for Venn diagram statistis. $P \leq 0.05$ was considered statistically significant. Error bars in figures represent s.d.

**Data availability.** The RNA-seq data in this study have been deposited in the National Center for Biotechnology Information (NCBI) Gene Expression Omnibus (GEO) under accession codes GSE98086 and GSE98507.

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

## Acknowledgements

We thank all members of the B Cell Biology Laboratory, M. Manzanares and C. Badía for helpful discussions; M. Busslinger for kindly providing the AID-CRE[+/TG] mice and critical reading of our manuscript; S. Mur for technical support; F. Sánchez-Cabo and M. Gómez for help with RNA-seq and statistical analyses; Genomics and Cellomics Units for technical advise; A. Losada, J. Mendez and V.G. de Yébenes for critical reading of the manuscript. NGS experiments were performed in the Genomics Unit of the CNIC. A.P.-G. was a fellow of the research training programme (FPU- AP2009-1732) funded by the Ministerio de Educación, Cultura y Deporte; E.M.-Z. is a fellow of the research training programme (FPI) funded by the Ministerio de Economía y Competitividad (BES-2014-069525); A.F.A.-P. and A.R.R. are supported by Centro Nacional de Investigaciones Cardiovaculares (CNIC). This work was funded by the research grant SAF2013-42767-R and SAF2016-75511-R (Plan Estatal de Investigación Científica y Técnica y de Innovación 2013–2016 Programa Estatal de I + D + i Orientada a los Retos de la Sociedad Retos Investigación: Proyectos I + D + i 2016, Ministerio de Economía, Industria y Competitividad) and co-funding by Fondo Europeo de Desarrollo Regional (FEDER) and the European Research Council Starting Grant programme (BCLYM-207844) to A.R.R. The CNIC is supported by the Ministry of Economy, Industry and Competitiveness (MEIC) and the Pro CNIC Foundation, and is a Severo Ochoa Center of Excellence (MEIC award SEV-2015-0505).

## Author contributions

A.P.-G., E.M.-Z. and A.R.R. designed experiments; N.G. provided the *Ctcf*-floxed mice strain; A.P.-G., E.M.-Z., A.F.A.-P. and J.M.L. performed experiments; A.P.-G., E.M.-Z. and A.R.R. analysed data and prepared figures, A.P.-G. and A.R.R. wrote the manuscript.

## Additional information

**Competing interests:** The authors declare no competing financial interests.

