## [Peer Review File · Nature Communications]

Reviewers' comments:

Reviewer #1 (Remarks to the Author):

The paper by Pérez-García et al addresses the role of CTCF on the germinal center development and transcriptional program of GC B cells.

The authors show that conditional knock out of Ctcf in mice results in complete abrogation of the GC response, both constitutive and induced by immunization with SRBC. Using an in vitro system they attempt to delineate the roles of Ctcf in maintaining a transcriptional program that prevents plasma cell differentiation. While the question posed is valid and interesting, there are several concerns that require to be addressed to validate the conclusions derived from these experiments.

Major comments:

While the phenotypic description looks solid, there is still limited novelty in the fact that complete knockout of Ctcf precludes development of GCs. Is there any cell type at all that is able to survive the complete absence of such a vital protein for chromosomal organization?

In all of the experiments the authors used Ctcf^{+/-} as the control. They should demonstrate that haploinsufficiency of Ctcf does not affect germinal center development to validate the use of these mice as controls.

All of the mechanistic aspects of the manuscript were done using an in vitro artificial system that would mimic GC reaction in vivo. It would be useful if the authors provide citations to refer to the original description of these protocols. In addition, the authors have to show that the difference in sensitivity to LPS/IL4 and T-cell/CD3/CD28 is not simply due to a differential proliferative capacity of the systems. In this sense, it would be possible that non cycling cells in LPS/IL4 setting would endure Ctcf loss, but cycling cells in the second system do not. Indeed, in Fig 3C, CD3/CD28 and actual GCs have a stronger activation of the cluster of cell cycle genes, as compared to LPS/IL4.

In the RNA-seq shown in Fig 4, it is not clear which cells the authors were using. As shown in Fig. 2E, the pool of hCD2 cells is reduced in Ctcf^{fl/fl}, presumably because they are dying after excision of Ctcf and posterior cycling. Therefore, the cells used in Fig 4 are either non excised or cells that are dying due to Ctcf deficiency. The authors should specify which time points they chose and show evidence that the cells were excised and viable at the time of RNA extraction.

The authors claim that Ctcf would repress Blimp-1 in GC cells and that would prevent GC exit and PC differentiation. However, they have not provided evidence of Ctcf being decreased in PCs. Is it possible to prevent PC differentiation by overexpression of Ctcf, such as for example, in cell lines?

Most importantly, the authors speculate that Ctcf would repress Blimp-1 without a mechanistic basis for such effect, and have not shown any evidence of Ctcf binding Blimp-1 locus, or modifying locus architecture in the CD3/CD28 system. The authors should show that the architecture of the locus is altered in Ctcf^{fl/fl} cells and prove that Blimp-1 is derepressed to support their conclusions.

Minor comments

In experiments addressing GC development in constitutive and induced GC by immunization there is a pool of remaining GCs after excision of the Ctcf allele. A safe assumption would be that those are unexcised cells. What is the efficiency of AID-cre to excise the floxed allele in these experiments?

IHC in Fig 1C. The authors should provide high quality images for revision and publication. Bcl6 IHC is

of a poor quality, showing staining of almost every cell in the spleen.

In the graphs in Fig 3A and 3B, the comparison should be made between the whole intersection of GCvsR and CD3vsR versus GCvsR and LPSvsR, instead of only the exclusive genes on those interaction gene sets.

It is not clear how the genes in Fig 3D were chosen or why they are relevant examples.

In Fig 4F, a more specific way to determine cell cycle should be used, such as BrdU/DAPI.

Reviewer #2 (Remarks to the Author):

This manuscript analyzes the outcome of genetic deletion of CTCF in germinal center B lymphocytes. The authors generate a mouse wherein a conditional allele of CTCF is deleted by CRE under control of the AID locus in a BAC transgene. They find decreases of GC B lymphocytes in both unimmunized and immunized animals.

The authors go on to assess the ability of these B lymphocytes to be activated under standard conditions in vitro. Surprisingly, they conclude that different activation pathways have differential requirements for CTCF.

Finally, the authors utilize RNAseq of in vitro and in vivo activated B cells to analyze gene expression dynamics and the impact of CTCF deletion. They conclude that CTCF is required for silencing of Blimp1 in the germinal center, leading to precocious plasma cell differentiation in cells in which CTCF is deleted.

The manuscript presents a reasonable descriptive account of how CTCF deletion impacts the GC reaction. The reader is left without substantive mechanistic insight into the process. Can class switching occur in the absence of CTCF? The authors suggest that CTCF directly regulates Blimp1 - is there mechanistic data to support this suggestion? Are the cell cycle genes perturbed direct targets of Blimp1? Does the XBP1 program proceed normally or not?

Reviewer #3 (Remarks to the Author):

The present manuscript investigates the consequences of the ablation of the CTCF-binding factor (CTCF) in germinal center (GC) B cells on the development of the GC response and GC-derived plasma cells (PC) by using a conditional *ctcf* allele and AID-Cre mice. It is found that deletion of CTCF strongly impairs the GC B-cell response. In vitro B-cell stimulation assays are employed to understand the mechanism behind the impairment of the GC response. RNA-seq analysis indicates that CTCF contributes to the maintenance of the GC B-cell transcriptional program and the high proliferation rate of GC B cells, and represses the expression of BLIMP1, a transcriptional repressor required for PC development. It is concluded from the data that CTCF has an essential role in orchestrating the maintenance of the GC transcriptional program by preventing PC differentiation.

The manuscript reports some definitely interesting observations on the role of CTCF in late B-cell development, but does not conclusively place CTCF in the network of the known transcription factors and epigenetic modulators that execute the initiation and the maintenance of the GC reaction, and does not reveal a specific function for CTCF in these processes. This is partly due to the circumstance that the background on which the rationale and the discussion of the results are based takes into consideration only the knowledge we had about the GC a decade ago. A lot of studies have been

published in the meantime that were not considered in the experimental setup and the data interpretation and that therefore, quite unfortunately, make this work somewhat untimely, because in the present form, it cannot provide a conclusive picture as to where and how in the GC response CTCF exerts its function. Several issues require attention in order to amend some incorrect statements and interpretation of the data in the context of the GC reaction as it has emerged from recent studies.

Major criticism:

(1) One major problem with the experimental approach is that mice heterozygous for the transgene were used as the control group. The investigators were lucky that they observed a phenotype, but the rationale as to why heterozygous mice were used as controls and not WT-AID-Cre mice, which is the correct control group, is unclear. While heterozygous mice have been used as controls in the distant past, it is now standard to use WT-Cre mice. This is not to say that the entire analysis should be repeated, but this reviewer is of the opinion that for a journal like Nature Communications, using heterozygous mice as controls is not acceptable because a gene in heterozygosity can be haploinsufficient.

(2) In the context of known literature about the GC reaction, the interpretation that CTCF is required for the maintenance of the GC reaction (Discussion page 14) cannot be made because the analysis time-point is day 7 after SRBC immunization, which is the approximately the day when the mature GC has been established. Thus, the interpretation would rather be that CTCF is required for the initiation of the GC reaction. It can also be required for the maintenance, but this is not experimentally addressed in the manuscript. Also the PP experiment cannot address establishment vs. maintenance. There is now ample literature on the different phases of the GC reaction, reviewed in Victora, Cell, 2014, De Silva et al. NRI 2015, and Mesin et al. 2016. This basis should be used to accordingly interpret the results in the context of the dynamics of the GC reaction.

(3) The introduction cites some incorrect papers. References Shaffer et al. and Klein et al. are incorrect to quote for somatic hypermutation, they make completely different points. Here, a review from Neuberger and perhaps an old Rajewsky review should be quoted after the entire sentence. Generally, the introduction is rather untimely, especially the exclusive focus on BCL6 and BLIMP1. Also in the discussion, sentences like '...the impact of chromatin structure changes during this reaction is poorly understood' is somewhat out of fashion in the light of work by the Melnick group.

(4) The statement that LPS/IL-4 activation is 'one of the gold standards for the study of GC biology' is not correct. LPS is an activator of the T-cell independent response. It is somewhat a mystery why LPS/IL-4 was used and not CD40/IL-4 which is generally considered a mimic of the T-cell dependent response. LPS/IL-4 is commonly used to study the function of genes of interest in class switching. The interpretation of the results shown in Fig. 3 would need to be revised accordingly, because it is not surprising that the LPS/IL-4 expression profile is so different from the others (T-independent vs. T-dependent response). The use of the CD3/CD28 activation assay and the data resulting from the comparison with GC B cells is very interesting, however.

REVIEWERS' COMMENTS:

Reviewer #2 (Remarks to the Author):

I thank the authors for a thoughtful and complete response to reviewer questions. The revisions provide mechanistic detail that bolsters the cellular data provided.

Reviewer #3 (Remarks to the Author):

Abstain from review of revision.

NCOMMS-16-24976
Point-by-point Reply to Reviewers' Comments
Almudena Ramiro

Reviewer #1 (Remarks to the Author):

The paper by Pérez-García et al addresses the role of CTCF on the germinal center development and transcriptional program of GC B cells.

The authors show that conditional knock out of Ctcf in mice results in complete abrogation of the GC response, both constitutive and induced by immunization with SRBC. Using an in vitro system they attempt to delineate the roles of Ctcf in maintaining a transcriptional program that prevents plasma cell differentiation. While the question posed is valid and interesting, there are several concerns that require to be addressed to validate the conclusions derived from these experiments.

We thank the reviewer for his/her thorough review of our manuscript and his/her positive assessment of our work. Please find below our detailed reply to the raised issues.

Major comments:

While the phenotypic description looks solid, there is still limited novelty in the fact that complete knockout of Ctcf precludes development of GCs. Is there any cell type at all that is able to survive the complete absence of such a vital protein for chromosomal organization?

We agree with the reviewer that CTCF is expectedly a critical, if not essential, protein for chromatin organization. This is why we were bewildered at not detecting any noticeable phenotype in CTCF deficient B cells stimulated in the presence of LPS and IL4. One could argue that these systems do not allow complete ablation of CTCF, but this is indeed the case for both (LPS/IL4 and T-CD3/CD28) stimulation settings (approximately 15% of CTCF protein remains in both cases), and susceptibility to CTCF loss in both systems is markedly different. On the other hand, incomplete ablation is a very common situation also for in vivo Cre models. Thus, our data provide an example of cells (LPS/IL4 stimulated B cells) that can tolerate CTCF loss (or very reduced levels of CTCF).

In addition, there is at least one example of insensitivity to CTCF loss in the literature (Ribeiro de Almeida et al, J Immunol 2009, reference #35 in the original version of our manuscript). In this work, the authors show that the proliferation defect found in CTCF deficient T cells could be bypassed with phorbol ester and ionomycin stimulation, which circumvents TCR signaling.

These two concepts are mentioned in the third and second paragraphs of the discussion, respectively.

In all of the experiments the authors used Ctcf^{+/-} as the control. They should demonstrate that haploinsufficiency of Ctcf does not affect germinal center development to validate the use of these mice as controls.

The reviewer is absolutely right that CTCF^{+/+} mice would have been a better control for all the experiments. The reason for not using them was that that involved generating groups of CTCF^{fl/fl}; AID-CreTG^{+/+} and CTCF^{+/+}; AID-

CreTG/+, and this only would have been possible by breeding CTCFfl/+; AID-CreTG/+ X CTCFfl/+; AID-Cre+/+, with an efficiency of only 12.5% per group. Most of the experiments in this work have been done using littermates, which would have been extremely difficult with the low efficiency of this breeding scheme. Instead, throughout the paper we used CTCFfl/fl; AID-CreTG/+ and CTCFfl/+; AID-CreTG/+ generated in CTCFfl/+; AID-CreTG/+ X CTCFfl/fl; AID-Cre+/+ breedings. Moreover, our data show that CTCF heterozygosity does not have a major phenotypic effect in germinal centers, and if a minor one existed, it would not invalidate our conclusions on the effect of CTCF loss.

In any case, we agree that the impact of CTCF heterozygosity is worth evaluating. Thus, we have analyzed GC formation in Peyer's patches and in immunization experiments in vivo in the following groups of mice, generated as described above:

1. CTCFfl/fl; AID-CreTG/+ CTCF deficient
2. CTCFfl/+; AID-CreTG/+ CTCF heterozygous, previously used as controls
3. CTCF+/+; AID-CreTG/+ CTCF wild type with Cre expression
4. CTCF+/+; AID-Cre+/+ CTCF wild type with Cre expression

Five-six mice of each group were analyzed for the presence of GC B cells, switched B cells and hCD2+ cells, as previously shown in Fig. 1A and 1B. We found that all these parameters were indistinguishable between groups 2, 3 and 4. We also did not detect histological alterations in CTCF heterozygous mice (group 2) when compared to CTCF wild type mice (groups 3 and 4) after PNA/DAPI/B220 staining and immunofluorescence detection. Thus, we conclude that CTCF heterozygosity does not detectably affect germinal center development, which validates their use as controls in all subsequent experiments. These new results have been included in **new Fig. 1A, B and C**.

All of the mechanistic aspects of the manuscript were done using an in vitro artificial system that would mimic GC reaction in vivo. It would be useful if the authors provide citations to refer to the original description of these protocols.

We agree that our in vitro experimental system is key in our study and we provide three citations that describe this protocol -references 28, 29 and 30 of our original manuscript:

Klaus, S.J. et al. Costimulation through CD28 enhances T cell-dependent B cell activation via CD40-CD40L interaction. *J Immunol* 152, 5643-5652 (1994).

Johnson-Leger, C., Christensen, J. & Klaus, G.G. CD28 co-stimulation stabilizes the expression of the CD40 ligand on T cells. *Int Immunol* 10, 1083- 1091 (1998).

Klaus, G.G., Holman, M., Johnson-Leger, C., Christenson, J.R. & Kehry, M.R. Interaction of B cells with activated T cells reduces the threshold for CD40-mediated B cell activation. *Int Immunol* 11, 71-79 (1999).

In addition, the authors have to show that the difference in sensitivity to LPS/IL4 and T-cell/CD3/CD28 is not simply due to a differential proliferative capacity of the systems. In this sense, it would be possible that non cycling cells in LPS/IL4 setting would endure Ctcf loss, but cycling cells in the second system do not. Indeed, in Fig

3C, CD3/CD28 and actual GCs have a stronger activation of the cluster of cell cycle genes, as compared to LPS/IL4.

The Reviewer raises an important point that also caught our mind in sight of the differential tolerance of B cells to CTCF loss in the two experimental conditions. Thus, we performed proliferation analysis in CFSE-loaded B cells stimulated either in the presence of LPS/IL4 or in the presence of CD3/CD28 stimulated T cells. To restrict the analysis to cells under comparable stimulation response, we gated on CD2⁺ (AID⁺) B cells. Interestingly, we found that B cells stimulated with CD3/CD28 T cells undergo less intense/slower cell division than LPS/IL4 stimulated B cells, as measured by dilution of CFSE cell division tracker. These results -now included in **new Fig. S2C**- indicate that the differential B cell susceptibility to CTCF loss under these stimulation conditions does not reflect a higher cell cycle demand of CD3/CD28 T stimulated B cells.

Regarding the Reviewer's remark on the seemingly stronger activation of the cluster of cell cycle genes in the CD3/CD28 T cell activation protocol, we would like to clarify the following: i) the "cell cycle" label on figure 3C refers to an enrichment analysis of the cell cycle pathway; we have now rephrased the figure legend description to avoid confusion; ii) only the 10% genes most differentially expressed between GC and resting B cells are shown in figure 3A, thus, other cell cycle related genes may and do behave in a different fashion; to clarify this point we have performed heat map analysis of all "cell cycle" related genes (\approx 480 genes, according to GO Enrichment Analysis, **Fig. R1**). This analysis reveals a complex pattern across the different stimulation conditions, whose impact in cell cycle regulation is difficult to predict; iii) finally, our gene enrichment analysis may include both positive and negative regulators of the cell cycle within complex regulatory networks, thus we would not like to imply that the expression pattern shown in Fig. 3A means a higher proliferation rate of CD3/CD28 activated B cells.

Finally, we provide additional evidence that, while CTCF deficiency decreases proliferation rate in CD3/CD28 stimulated B cells (Fig. 5A), it does not noticeably affect proliferation of LPS/IL4 stimulated B cells (**new Fig. S3**).

Fig. R1. Heatmap analysis of RNAseq data of cell cycle-related genes. Cell cycle genes (484) were chosen using GO Enrichment Analysis. Z-scores for resting, GC, LPS/IL4 and CD3/D28 stimulated B cells are represented. Clustering was performed using the average linkage method based on Pearson correlation distance. Each column shows an independent replicate.

In the RNA-seq shown in Fig 4, it is not clear which cells the authors were using. As shown in Fig. 2E, the pool of hCD2 cells is reduced in *Ctcf^{fl/fl}*, presumably because they are dying after excision of *Ctcf* and posterior cycling. Therefore, the cells used in Fig 4 are either non excised or cells that are dying due to *Ctcf* deficiency. The authors should specify which time points they chose and show evidence that the cells were excised and viable at the time of RNA extraction.

We apologize if the information provided on cell sorting was not sufficient. In Fig. 4 we are showing different aspects of our RNAseq experiments. We compare CTCF deficient (*CTCF^{fl/fl} AIDCreTG/+*) with control (*CTCF^{fl/+} AIDCreTG/+*) cells after 48 hours of stimulation either in the presence of LPS/IL4 (Fig. 4A and 4B) or in the presence of CD3/CD28 and T cells (Fig. 4A-F). In order to select stimulated cells within cultures, we sort for hCD2⁺, B220⁺ cells, dead cells are excluded by DAPI staining. The sorting strategy and post-sort analysis is included as **Fig. R2** (representative plots of a pool 3 CD3/CD28 T:B cell cultures). This information (DAPI⁻, B220⁺, hCD2⁺ cells, sorted after 48 hours of stimulation) is now specified in the figure legend.

Fig. R2. Gating strategy and post-sort analysis for selection of B cells used in RNAseq experiments. CTCF^{fl/fl};AIDCreTG/+ and CTCF^{fl/+};AIDCreTG/+ B cells stimulated for 48h hours either in the presence of LPS/IL4 or in the presence of CD3/CD28 T stimulation were sorted as DAPI-, B220+, hCD2+ cells. Representative plots of B cells stimulated with CD3/CD28 T cells are shown.

In addition, we have never found evidence for a survival defect in CTCF^{fl/fl} AIDcreTG/+ B cells in either condition in vitro, suggesting that if it existed, it should have a minor impact on the cell recovery for these experiments. Quantification of caspase 3 staining in both genotypes and both stimulation conditions is shown in **Fig. R3** to illustrate this point.

Fig. R3. Measurement of cell death in CTCF^{fl/fl};AIDCreTG/+ and CTCF^{fl/+};AIDCreTG/+ B cells stimulated for 48h hours either in the presence of LPS/IL4 or in the presence of CD3/CD28 T stimulation. Histograms show FACS analysis of active Caspase 3 staining after gating in live (DAPI-), activated (hCD2+) cells. Graphs on the right show geometric mean fluorescence intensity (MFI), error bars indicate standard deviation (n=3).

Finally, we show that under these experimental conditions (48 hours of stimulation, DAPI-, B220+, hCD2+) only 15% of CTCF protein remains in the CD3/CD28 T cultures (Fig. 2G), similar to the CTCF remaining amount in LPS/IL4 cultures (Fig. 2C), i.e. where we have not detected any deleterious effect of CTCF deficiency. Moreover, we have now set up the conditions to quantify the excision of the CTCFfl allele (**new Fig S2A**) by qPCR of genomic DNA in CTCFfl/fl and CTCFfl/+ DAPI-, B220+, hCD2+ sorted B cells. We find that CTCFfl/fl B cells yielded twice as much amplification (i.e. twice as many excised alleles) than CTCFfl/+ B cells (**new Fig. S2B**). Sorted DAPI-, B220+, hCD2- as well as B cells from CTCF+/+ mice are included as controls. We believe that together these data provide strong evidence that our sort protocol isolates viable, excised cells rather than non excised or cells that are dying due to CTCF deficiency.

The authors claim that *Ctcf* would repress Blimp-1 in GC cells and that would prevent GC exit and PC differentiation. However, they have not provided evidence of *Ctcf* being decreased in PCs. Is it possible to prevent PC differentiation by overexpression of *Ctcf*, such as for example, in cell lines?

The Reviewer raises a very interesting point. We have addressed Blimp-1 regulation during GC to PC differentiation by analyzing the RNAseq data reported by the Nutt lab (Shi et al, Nat Immunol 16, 663). We found that indeed CTCF expression drops at the transition from GC to PC differentiation. These data are now included as **new Fig. 5D**.

Regarding the impact of CTCF overexpression on PC differentiation, we have tried to approach this issue but have come across several difficulties. First, there is not a well-accepted cell line protocol to study PC differentiation in vitro; thus we decided to use primary spleen B cells stimulated with LPS. Under these conditions B cells proliferate, express AID, switch to IgG3 and end up expressing CD138 and secreting immunoglobulin (see, for instance, Saini et al., Nat Immunol 15, 275), features that partially recapitulate the PC differentiation.

Thus, to address the effect of CTCF overexpression on PC differentiation, we cloned CTCF cDNA into the pMXPIE retroviral vector -where GFP can be used to track transduction, Barreto et al Mol Cell 12, 501-, and transduced LPS-stimulated B cells. B cells transduced with pMXPIE empty vector were used as controls. We repeated the experiment with 6 mice, and we invariably found that CTCF-transduced (GFP+) cells were hardly detectable (<0.2%), while we detected 5-15% GFP+ control transduced cells (**Fig. R4**, plots on the left). Moreover, the proportion of hCD2+ cells -hence, cells that express AID- is decreased within the GFP+ CTCF+ fraction (lower right plot) when compared to the proportion found within GFP+ control cells (upper right plot), suggesting that CTCF overexpression could be deleterious for activated B cells. In turn, this observation could be reminiscent of data reported by the Morse lab, where CTCF heterologous expression in the WEHI cell line inhibited cell growth and induced cell death (Qi et al., Proc Natl Acad Sci USA 100, 633). One could speculate that excessive CTCF may promote illegitimate chromatin loops with an unforeseeable functional impact, in this case compromised cell survival. However, given the extremely low number of CTCF transduced cells we cannot unequivocally ascertain this point, at least with this experimental system. Thus, we believe it safer to leave this information out of the manuscript data and provide it solely for the Reviewers' information.

Here Fig. R4. was redacted

Most importantly, the authors speculate that Ctcf would repress Blimp-1 without a mechanistic basis for such effect, and have not shown any evidence of Ctcf binding Blimp-1 locus, or modifying locus architecture in the CD3/CD28 system. The authors should show that the architecture of the locus is altered in Ctcffl/fl cells and prove that Blimp-1 is derepressed to support their conclusions.

We agree with the Reviewer in that it is important to show evidence in this regard. To that end, we have analyzed available ChIP-Seq data corresponding to CTCF IP from primary mouse B cells stimulated in the presence of LPS and IL4 (GSE33819 and Nakahashi et al, Cell Rep 3, 1678). We found three distinct peaks (PeakA-C) of CTCF binding to the Prdm1 locus (data are now included as **new Fig. 5F**). To assess whether CTCF binding also occurs under our specific stimulation conditions, we stimulated B cells in the presence of CD3/CD28 and T cells, and performed ChIP-qPCR analysis of the three regions (PeakA-C) (**new Fig. 5F**). H19 regions 1, 2 and 3 were used as positive controls of CTCF binding (Hark et al Nature 405, 486), and IP with an isotypic control was used as a negative control. We found that indeed CTCF binds to Blimp1 at regions A, B and C in B cells stimulated with CD3/CD28 and T cells (**new Fig. 5G**).

Derepression (i.e. increased expression) of Blimp1 in CTCFfl/fl cells is shown in Fig. 5E.

Minor comments

In experiments addressing GC development in constitutive and induced GC by immunization there is a pool of remaining GCs after excision of the Ctcf allele. A safe assumption would be that those are unexcised cells. What is the efficiency of AID-cre to excise the floxed allele in these experiments?

We agree with the reviewer that there are a few GC cells in CTCFfl/fl mice that may have escaped Cre-mediated deletion. However, their extremely low numbers preclude any molecular analysis, including excision assessment. CTCF excision has been measured in vitro as suggested by the Reviewer (**new Fig. S2B**)

IHC in Fig 1C. The authors should provide high quality images for revision and publication. Bcl6 IHC is of a poor quality, showing staining of almost every cell in the spleen.

We thank the reviewer for pointing this out. We have now replaced the original Bcl6 IHC images with new immunofluorescence images of spleens from immunized mice stained with DAPI, B220 and PNA (**new Fig. 1C**). We believe that the new micrographs display a much better image quality of the GCs and of the difference across genotypes.

In the graphs in Fig 3A and 3B, the comparison should be made between the whole intersection of GCvsR and CD3vsR versus GCvsR and LPSvsR, instead of only the exclusive genes on those interaction gene sets.

Following the Reviewer's suggestion we have now represented the whole overlapping genes (highlighted with white dotted lines in the Venn diagrams) in the bar graphs of Fig. 3A and 3B.

It is not clear how the genes in Fig 3D were chosen or why they are relevant examples.

Genes were chosen based on their functional relevance, mostly in cell cycle (for instance, Cdk1, Cdk6, Cdc25a, among others) and B cell differentiation or function (Blimp1, Tnfrsf13b, Fcer2a). This is now specified in figure legend 3.

In Fig 4F, a more specific way to determine cell cycle should be used, such as BrdU/DAPI.

In addition to regulation of cell cycle genes shown in Fig. 4F, direct cell cycle analysis by Hoescht staining is shown in Fig. 5A.

Reviewer #2 (Remarks to the Author):

This manuscript analyzes the outcome of genetic deletion of CTCF in germinal center B lymphocytes. The authors generate a mouse wherein a conditional allele of CTCF is deleted by CRE under control of the AID locus in a BAC transgene. They find decreases of GC B lymphocytes in both unimmunized and immunized animals. The authors go on to assess the ability of these B lymphocytes to be acytivated under standard conditions in vitro. surprisingly, they conclude that different activation pathways have differential requirements for CTCF. Finally, the authors utilize RNAseq of in vitro and in vivo activated B cells to analyze gene expression dynamics and the impact of CTCF deletion. They conclude that CTCF is required for silencing of Blimp1 in the germinal center, leading to precocious plasma cell differentiation in cells in which CTCF is deleted.

The manuscript presents a reasonable descriptive account of how CTCF deletion impacts the GC reaction. The reader is left without substantive mechanistic insight into the process.

We are thankful for the Reviewer's insightful comments and suggestions. Please find below the detailed answer to all the issues raised by the Reviewer.

Can Class switching occur in the absence of CTCF?

This is an interesting point. We have found that CTCF^{fl/fl} cells stimulated in the presence of LPS and IL4 indeed can switch to IgG1 with a similar efficiency than control littermates. This result is provided as **Fig. R5**. In the model of CD3/CD28 T stimulation, the yield of switched cells is poor and thus we could not address this point reliably. We dare to hypothesize that the defect in proliferation could be accompanied by a defect in class switch recombination, but a different experimental model would be required to assess this unequivocally.

Here **Fig. R5**. was redacted

The authors suggest that CTCF directly regulates Blimp1 - is there mechanistic data to support this suggestion?

We now provide new evidence that CTCF binds the Prdm1 (Blimp-1) gene (**new Fig. 5F and G**). Specifically, we analyzed CTCF ChIP-Seq available data on LPS/IL4 stimulated B cells and found that there are three clear peaks of CTCF binding at three regions of Prdm1 (PeaksA-C) (Fig. 5F). In addition, we performed ChIP-qPCR experiments on B cells stimulated with CD3/CD28 and T cells and found that in our experimental conditions CTCF also binds Prdm1 at Peaks A-C. H19 regions were used as positive control for CTCF binding and isotype-matched IgG IP was used as a negative control (Fig. 5G). This new data provides evidence for direct interaction of CTCF with the Blimp-1 gene and suggests the mechanistic basis for the observed phenotype.

Are the cell cycle genes perturbed direct targets of blimp1?

To address this issue we have analyzed previously reported Blimp-1 ChIPseq data on plasmablasts (Minnich et al., Nat Immunol 17, 331) and we found that many of the

deregulated cell cycle genes can be direct targets of Blimp-1, as measured by direct binding. These data is shown in **new Fig. 5I**. We also find that other deregulated cell cycle genes are not direct targets of Blimp-1, but this could well be explained by indirect effect of Blimp-1 through another regulator, or by additional, Blimp-1 independent effects of CTCF, as mentioned in the discussion.

Does the XBPI program proceed normally or not?

At the transition to antibody-secreting cell the unfolding protein response (UPR), needs to be activated to allow proper Ig folding and secretion. Blimp-1 is important role in this process by directly regulating Atf6 which in turn promotes the Xbp1 expression. Although we have not found alterations in Xbp1 expression in the absence of CTCF, we did detect a deregulation of UPR genes, including an increase in Atf6 levels.

Reviewer #3 (Remarks to the Author):

The present manuscript investigates the consequences of the ablation of the CCCTC-binding factor (CTCF) in germinal center (GC) B cells on the development of the GC response and GC-derived plasma cells (PC) by using a conditional *ctcf* allele and AID-Cre mice. It is found that deletion of CTCF strongly impairs the GC B-cell response. In vitro B-cell stimulation assays are employed to understand the mechanism behind the impairment of the GC response. RNA-seq analysis indicates that CTCF contributes to the maintenance of the GC B-cell transcriptional program and the high proliferation rate of GC B cells, and represses the expression of BLIMPI, a transcriptional repressor required for PC development. It is concluded from the data that CTCF has an essential role in orchestrating the maintenance of the GC transcriptional program by preventing PC differentiation.

The manuscript reports some definitely interesting observations on the role of CTCF in late B-cell development, but does not conclusively place CTCF in the network of the known transcription factors and epigenetic modulators that execute the initiation and the maintenance of the GC reaction, and does not reveal a specific function for CTCF in these processes. This is partly due to the circumstance that the background on which the rationale and the discussion of the results are based takes into consideration only the knowledge we had about the GC a decade ago. A lot of studies have been published in the meantime that were not considered in the experimental setup and the data interpretation and that therefore, quite unfortunately, make this work somewhat untimely, because in the present form, it cannot provide a conclusive picture as to where and how in the GC response CTCF exerts its function. Several issues require attention in order to amend some incorrect statements and interpretation of the data in the context of the GC reaction as it has emerged from recent studies.

We thank the reviewer for finding our manuscript interesting and for his/her constructive comments, which are addressed in detail below.

Major criticism:

(1) One major problem with the experimental approach is that mice heterozygous for the transgene were used as the control group. The investigators were lucky that they observed a phenotype, but the rationale as to why heterozygous mice were used as controls and not WT-AID-Cre mice, which is the correct control group, is unclear. While heterozygous mice have been used as controls in the distant past, it is now standard to use WT-Cre mice. This is not to say that the entire analysis should be repeated, but this reviewer is of the opinion that for a journal like Nature Communications, using heterozygous mice as controls is not acceptable because a gene in heterozygosity can be haploinsufficient.

We agree with the reviewer that CTCF^{+/+} mice would have been a better control for all the experiments. However, breeding of CTCF^{fl/+}; AID-Cre^{TG/+} X CTCF^{fl/+}; AID-Cre^{+/+} mice generates CTCF^{fl/fl}; AID-Cre^{TG/+} and CTCF^{+/+}; AID-Cre^{TG/+} mice with only 12.5% efficiency, which makes working with littermates extremely difficult. This is why we decided to use CTCF^{fl/fl}; AID-Cre^{TG/+} and CTCF^{fl/+}; AID-Cre^{TG/+} generated in CTCF^{fl/+}; AID-Cre^{TG/+} X CTCF^{fl/fl}; AID-Cre^{+/+} breedings.

Following the Reviewer's suggestion we have now evaluated the phenotype of CTCF heterozygote mice in immunization experiments in vivo in the following groups of mice, generated by CTCF^{fl/+}; AID-Cre^{TG/+} X CTCF^{fl/+}; AID-Cre^{+/+} breedings:

1. CTCF^{fl/fl}; AID-Cre^{TG/+} CTCF deficient
2. CTCF^{fl/+}; AID-Cre^{TG/+} CTCF heterozygous, previously used as controls
3. CTCF^{+/+}; AID-Cre^{TG/+} CTCF wild type with Cre expression
4. CTCF^{+/+}; AID-Cre^{+/+} CTCF wild type with Cre expression

Mice were immunized and analyzed for the presence of GC B cells, switched B cells and hCD2⁺ cells, as previously shown in the original Fig. 1A and 1B. We found that all these parameters were indistinguishable between groups 2, 3 and 4. We also did not detect histological alterations in CTCF heterozygous mice (group 2) when compared to CTCF wild type mice (groups 3 and 4) after PNA/DAPI/B220 staining and immunofluorescence detection. Thus, we conclude that heterozygosity of CTCF does not detectably affect germinal center development, which validates their use as controls in all subsequent experiments. These new results have been included in Fig. 1A, B and C.

(2) In the context of known literature about the GC reaction, the interpretation that CTCF is required for the maintenance of the GC reaction (Discussion page 14) cannot be made because the analysis time-point is day 7 after SRBC immunization, which is the approximately the day when the mature GC has been established. Thus, the interpretation would rather be that CTCF is required for the initiation of the GC reaction. It can also be required for the maintenance, but this is not experimentally addressed in the manuscript. Also the PP experiment cannot address establishment vs. maintenance. There is now ample literature on the different phases of the GC reaction, reviewed in Victora, Cell, 2014, De Silva et al. NRI 2015, and Mesin et al. 2016. This basis should be used to accordingly interpret the results in the context of the dynamics of the GC reaction.

We apologize if we failed to make our point clear enough. We are aware that GCs have an early phase of expansion at days 5 to 7 and that later on, in mature GCs,

light zone dark zone are formed and iterative cycles of selection shape affinity maturation (work mostly from Gabriel Victora but also from other labs). However in our manuscript we never intended to refer to early GC but rather we use the term “initiation of the GC transcriptional program”. The reason why we make this point refers to the fact that the expression of AID is considered one of the hallmarks of the GC transcriptional program and that in our model CTCF deletion is triggered by AID expression. Thus, in our model, by the time of the initiation of the GC transcriptional program (i.e. as defined by AID expression), B cells still express CTCF and its deletion is necessarily delayed (because of the time it takes for Cre to delete the floxed allele, and for CTCF protein to disappear) with respect to the initiation of the GC transcriptional program. This is why we make the point in our discussion that “While it is likely that CTCF is required for GC initiation, the delay of CTCF depletion –dependent on AID expression- in our model is not suitable to specifically ascertain that point”, in the sense that rigorously speaking, we could only test initiation of the transcriptional program in a model where CTCF is already absent by the time AID (as a marker of the GC program) is expressed. Accordingly, we use the term “maintenance of the GC transcriptional program” in opposition to “initiation” but we do not refer at all to mature GC and the process of affinity maturation, which we did not and could not approach with these tools. We apologize again if the distinction between initiation-maintenance of GC transcriptional program seemed to overlap with the definition early-mature GCs (as defined for instance in Victora Cell 2014). We have now tried to clarify this further in our manuscript and have removed the word “maintains” from the title to avoid confusion.

(3) The introduction cites some incorrect papers. References Shaffer et al. and Klein et al. are incorrect to quote for somatic hypermutation, they make completely different points. Here, a review from Neuberger and perhaps an old Rajewsky review should be quoted after the entire sentence. Generally, the introduction is rather untimely, especially the exclusive focus on BCL6 and BLIMP1. Also in the discussion, sentences like ‘...the impact of chromatin structure changes during this reaction is poorly understood’ is somewhat out of fashion in the light of work by the Melnick group.

We apologize for our involuntary mistake at citing the wrong references for SHM. We have now cited appropriate references after that sentence and other reference mistakes have also been fixed. We apologize also for having missed to cite the Bunting et al Immunity paper that came out a few days before the submission of our manuscript and we failed to timely include.

(4) The statement that LPS/IL-4 activation is ‘one of the gold standards for the study of GC biology’ is not correct. LPS is an activator of the T-cell independent response. It is somewhat a mystery why LPS/IL-4 was used and not CD40/IL-4 which is generally considered a mimic of the T-cell dependent response. LPS/IL-4 is commonly used to study the function of genes of interest in class switching. The interpretation of the results shown in Fig. 3 would need to be revised accordingly, because it is not surprising that the LPS/IL-4 expression profile is so different from the others (T-independent vs. T-dependent response). The use of the CD3/CD28

activation assay and the data resulting from the comparison with GC B cells is very interesting, however.

We have now rephrased the sentence “LPS/IL-4 activation is one of the gold standards for the study of GC biology”. Regarding the activation with CD40/IL4, we indeed tried that and other similar activation protocols, and in none of them did the absence of CTCF promote a detectable phenotype (in terms of viability, cell proliferation, CD2 expression or class switch recombination). We decided to stick with LPS/IL4 for our manuscript because it provides a good model for class switch and because numerous genomewide transcriptomic and ChIPSeq data are already publicly available for this stimulation. We are very glad that the reviewer finds the CD3/CD28 assay and the GC comparison interesting.